# Structures and transport dynamics of a *Campylobacter jejuni* multidrug efflux pump

Chih-Chia Su[1], Linxiang Yin[2], Nitin Kumar[3], Lei Dai[4], Abhijith Radhakrishnan[3], Jani Reddy Bolla[3], Hsiang-Ting Lei[3], Tsung-Han Chou[1], Jared A. Delmar[1], Kanagalaghatta R. Rajashankar[5], Qijing Zhang[4], Yeon-Kyun Shin[2] & Edward W. Yu[1,3]

Resistance-nodulation-cell division efflux pumps are integral membrane proteins that catalyze the export of substrates across cell membranes. Within the hydrophobe-amphiphile efflux subfamily, these resistance-nodulation-cell division proteins largely form trimeric efflux pumps. The drug efflux process has been proposed to entail a synchronized motion between subunits of the trimer to advance the transport cycle, leading to the extrusion of drug molecules. Here we use X-ray crystallography and single-molecule fluorescence resonance energy transfer imaging to elucidate the structures and functional dynamics of the *Campylobacter jejuni* CmeB multidrug efflux pump. We find that the CmeB trimer displays a very unique conformation. A direct observation of transport dynamics in individual CmeB trimers embedded in membrane vesicles indicates that each CmeB subunit undergoes conformational transitions uncoordinated and independent of each other. On the basis of our findings and analyses, we propose a model for transport mechanism where CmeB protomers function independently within the trimer.

[1] Department of Physics and Astronomy, Iowa State University, Ames, IA 50011, USA. [2] Roy J. Carver Department of Biochemistry, Biophysics and Molecular Biology, Iowa State University, Ames, IA 50011, USA. [3] Department of Chemistry, Iowa State University, Ames, IA 50011, USA. [4] Department of Veterinary Microbiology, College of Veterinary Medicine, Iowa State University, Ames, IA 50011, USA. [5] NE-CAT and Department of Chemistry and Chemical Biology, Cornell University, Argonne National Laboratory, Bldg. 436E, 9700 S. Cass Avenue, Argonne, IL 60439, USA. Chih-Chia Su, Linxiang Yin and Nitin Kumar contributed equally to this work. Correspondence and requests for materials should be addressed to E.W.Y. (email: ewyu@iastate.edu)

Campylobacter jejuni is responsible for more than 400 million cases of human enterocolitis each year worldwide[1, 2]. This infection is capable of triggering an autoimmune response and initiating the development of Guillain-Barre syndrome[2]. C. jejuni is frequently found in the intestinal tracts of animals and can be spread to humans through contaminated food, water, or raw milk. Fluoroquinolone and macrolide antibiotics are commonly used as treatment strategies for human campylobacteriosis[3]. However, Campylobacter has become resistant to these antimicrobials[4–6]. Recently, the Centers for Disease Control and Prevention have listed drug-resistant Campylobacter as a serious threat in the United States. Resistance of Campylobacter to antibiotics is mediated by multiple mechanisms[5, 7] and multidrug efflux is one of the major causes of failure of drug-based treatments. Multidrug efflux pumps contribute significantly to both intrinsic and acquired resistance to various antimicrobials in C. jejuni. The resistance compromises the effectiveness of clinical therapy and affects the duration of clinical treatment. It has been suggested that acquisition of fluoroquinolone resistance provides a fitness advantage on C. jejuni in the animal host[8]. Needless to say, new therapeutic strategies are needed to develop to strive antibiotic resistant Campylobacter.

The best characterized multidrug efflux system in C. jejuni is the Cme (Campylobacter multidrug efflux) tripartite system[9–11].

The Cme locus consists of three tandemly linked genes (cmeABC) encoding protein components of the tripartite Cme efflux pump (CmeA, CmeB, and CmeC), where all three components are absolutely required for substrate expulsion. This tripartite system is composed of the CmeB efflux pump, an inner membrane resistance-nodulation-cell division (RND)[12] transport protein that contains substrate-binding sites and transduces the electrochemical energy required for pumping drugs out of the cell; the CmeA periplasmic protein, a member of the membrane fusion protein family; and the CmeC outer membrane-associated protein that is integral to the outer membrane. Mutations on this tripartite system have been found to have a drastic effect on drug susceptibility[9, 13].

To understand the transport mechanism of the CmeB efflux pump from C. jejuni, we here define the X-ray structures of this membrane protein, which assembles as a trimer. Using single-molecule fluorescence resonance energy transfer (sm-FRET) imaging, we demonstrate that each CmeB protomer within the trimer is able to function independently.

## Results

**Structures of C. jejuni CmeB.** Two distinct conformations of CmeB with space groups C2 (form I) and P1 (form II) were captured in two different forms of crystals (Fig. 1, Table 1 and Supplementary Figs. 1 and 2). Overall, CmeB adopts the fold of a

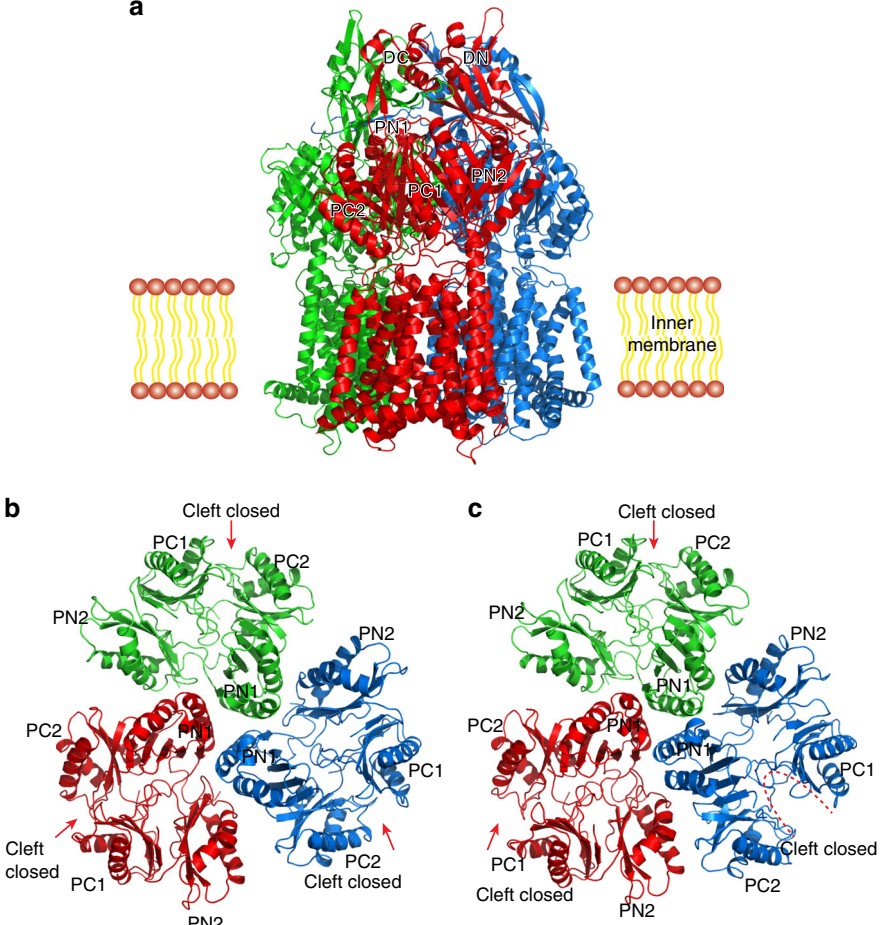

**Fig. 1** Structure of the CmeB efflux pump. **a** Ribbon diagram of the form I structure of the CmeB homotrimer viewed in the membrane plane. Each subunit of CmeB is labeled with a different color. Subdomains DN, DC, PN1, PN2, PC1, and PC2 are labeled on the front protomer (red). **b** Top view of the form I CmeB trimer. Each subunit of CmeB is colored differently. The six subdomains DN, DC, PN1, PN2, PC1, and PC2 are labeled. In this conformation, the periplasmic cleft between PC1 and PC2 is closed in each protomer. **c** Top view of the form II CmeB trimer. In this conformation, only one out of the three periplasmic clefts formed between PC1 and PC2 is open

**Table 1 Data collection and refinement statistics**

|  | CmeB (form I) (5T0O) | CmeB (form II) (5LQ3) |
|---|---|---|
| *Data collection* | | |
| Space group | C2 | P1 |
| Cell dimensions | | |
| *a, b, c* (Å) | 300.71, 147.54, 120.03 | 120.85, 127.94, 168.58 |
| *α, β, γ* (°) | 90.0, 90.9, 90.0 | 99.8, 99.4, 85.0 |
| Resolution (Å) | 50–3.15 (3.26–3.15)ᵃ | 50–3.55 (3.68–3.55) |
| $R_{pim}$ | 9.5 (45.5) | 11.4 (76.9) |
| *I*/σ(*I*) | 14.6 (2.9) | 8.12 (1.4) |
| $CC_{1/2}$ | (0.57) | (0.49) |
| Completeness (%) | 99.9 (99.8) | 91.3 (94.1) |
| Redundancy | 3.7 (3.6) | 3.7 (3.7) |
| | | |
| *Refinement* | | |
| Resolution (Å) | 50–3.15 | 50–3.55 |
| No. reflections | 75,644 | 104,139 |
| $R_{work}$/$R_{free}$ | 21.09/26.78 | 22.63/26.77 |
| No. atoms | | |
| Protein | 23,896 | 47,787 |
| *B* factors | | |
| Protein | 69.19 | 100.92 |
| R.m.s. deviations | | |
| Bond lengths (Å) | 0.009 | 0.002 |
| Bond angles (°) | 1.39 | 0.52 |
| Ramachandran plot | | |
| Favored (%) | 93 | 92 |
| Allowed (%) | 6 | 7 |
| Outliers (%) | 1 | 1 |

ᵃValues in parentheses are for highest-resolution shell.

typical hydrophobe-amphiphile efflux (HAE)-RND-type protein and forms a homotrimer, with its pseudo threefold symmetrical axis positioning perpendicular to the membrane surface. Thus far, all known HAE-RND pumps are found to be trimeric in form, suggesting that this may be the most stable oligomerization state for these membrane proteins. Each subunit of CmeB contains 12 transmembrane helices (TM1-TM12) and a large periplasmic domain created by two extracellular loops between TM1 and TM2, and between TM7 and TM8, respectively. This periplasmic domain can be divided into six subdomains: PN1, PN2, PC1, PC2, DN, and DC (Fig. 1a). Subdomains PN1, PN2, PC1, and PC2 form the pore domain, with PN1 making up the central pore and stabilizing the trimeric organization. Subdomains DN and DC, however, contribute to form the docking domain of the pump.

A cleft is formed between subdomains PC1 and PC2. Presumably, this cleft creates an entrance, allowing substrates to move into the pump via the periplasm. Deep inside the cleft, the CmeB pump forms a large internal cavity (Supplementary Fig. 3a). In AcrB, this cavity has been shown to form an important binding site (Supplementary Fig. 3b), which plays a predominant role in recognizing substrates for export[14–16]. Recently, a drug resistance-enhanced variant of CmeB has been identified in clinical isolates of *C. jejuni*. This mutant pump is able to confer high-level bacterial resistance to multiple antibiotics, including chloramphenicol, ciprofloxacin, erythromycin, and tetracycline[16]. Interestingly, 22 mutated residues are found to localize within this drug-binding cavity. The corresponding amino acids of many of these mutated residues, including M607L, A152D, T88Q, M292I, and M571L, have been observed to be critical for multidrug binding in AcrB[14, 15].

The structures of AcrB indicate that this multidrug efflux pump is capable of forming an asymmetric trimer, in which the three protomers are distinct and display different conformational

states ("access", "binding", and "extrusion")[14, 17, 18]. This structural dissimilarity has led to a proposed transport mechanism that the three protomers of an RND transporter must cooperate and synchronize to go through these three different states to export drugs. In both the "access" and "binding" protomers of AcrB, the periplasmic cleft created by subdomains PC1 and PC2 are open. However, this cleft is closed in the "extrusion" protomer. Thus, the asymmetric trimer of AcrB features a conformational state of two periplasmic clefts open and one cleft closed. During drug binding and extrusion, it was proposed that the three periplasmic clefts within the trimer have to open and close accordingly in order to advance the transport cycle.

Both crystal structures of CmeB depict that this pump also forms an asymmetric trimer. In the form I structure, the three periplasmic clefts of the CmeB trimer are closed (Fig. 1b). Although the conformations of the three monomers are different from each other, they are more similar to the "extrusion" form of AcrB (Supplementary Table 1). A channel for extrusion is found in each protomer of our form I structure (Fig. 2a, b). We therefore assigned the conformational state of these three protomers as the "extrusion" form.

Surprisingly, in the form II structure, the periplasmic cleft of one of the protomers is open (Fig. 1c), albeit similar to the conformation of the "binding" state of AcrB (Supplementary Table 2). An elongated channel is found in the periplasmic domain of this protomer. It was observed that this channel leads through the opening of the periplasmic cleft, exposed to solvent in the periplasm (Fig. 2c, d). We labeled this conformer as the "binding" form of CmeB. However, the periplasmic clefts of the other two protomers are closed in conformation. One of these two CmeB molecules is more related to the structure of the "extrusion" conformers of the form I structure. Like the three protomers of the form I structure, the periplasmic domain of this conformer also creates an extrusion channel (Fig. 2c, d and Supplementary Fig. 4). Thus, this conformer was classified as the "extrusion" form. All of the extrusion protomers in both form I and form II crystals share very similar structural features, which include the conformation of transmembrane helices (Supplementary Table 3 and Supplementary Fig. 5a) and position of side chains of residues within the proton-relay network such as D409 and D410 of TM4 and K935 of TM10 (Supplementary Fig. 5b). Interestingly, the other CmeB molecule of the form II structure displays a distinct conformation, forming a new state that is different from the "extrusion" form. No channel was found in this conformer (Fig. 2c, d and Supplementary Fig. 4c). This conformation probably represents one of the intermediates that the CmeB pump must go through during the transport cycle. However, the conformation of this protomer is similar to the "resting" state of apo-CusA[19, 20], a specific heavy-metal RND efflux pump that recognizes Cu(I) and Ag(I) ions. A similar conformational state in the AcrB pump has also been found based on molecular dynamics simulations[21]. We thus designated this conformation as the "resting" state of the CmeB efflux pump.

Our structures of CmeB do not show the typical conformation of an asymmetric trimer with two periplasmic clefts open and one cleft closed (Supplementary Fig. 6). We then went through the existing crystal structures of these RND proteins, including AcrB[14, 15, 22–24], MexB[25], MtrD[26], and CusA[19, 27], available in the protein data bank. We found that the trimer can have three periplasmic clefts open or closed at a time, in addition to the asymmetric conformation of the AcrB trimer (Supplementary Fig. 7). On the basis of this structural information, we postulated that individual protomers of these trimeric RND pumps could bind and export substrates independently instead of operating in a synchronized fashion. Thus, each protomer may autonomously

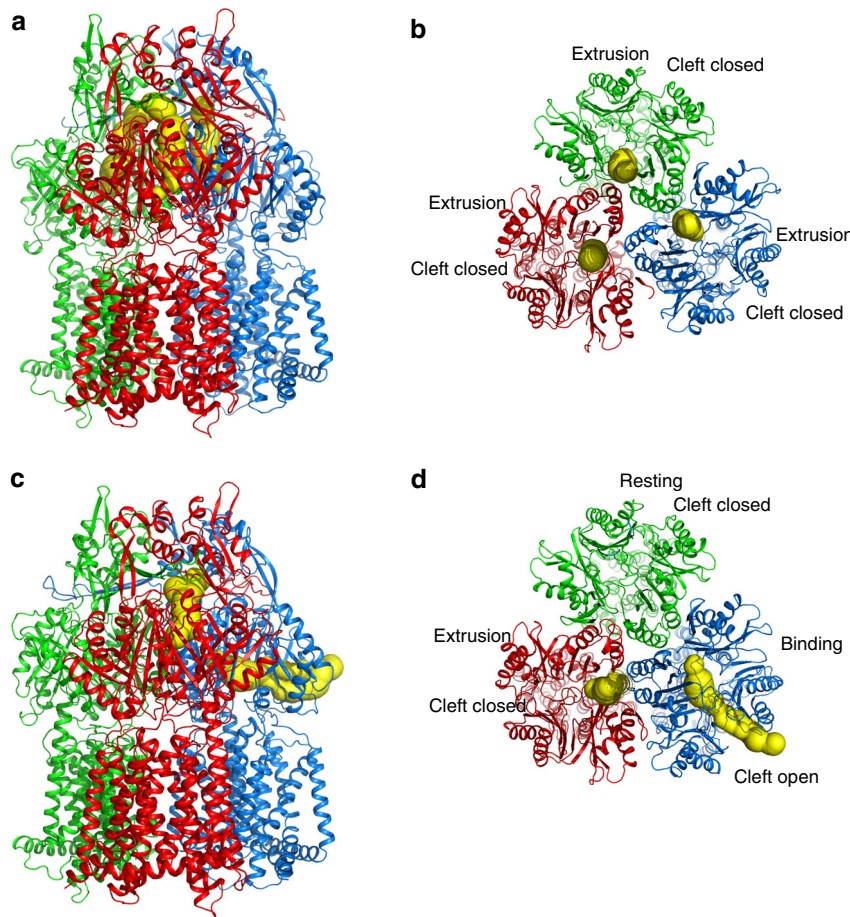

**Fig. 2** Channels in the CmeB pump. **a** Side view and **b** top view of the channels formed by the periplasmic domain of the form I structure of the CmeB trimer. Each protomer creates an extrusion channel for drug export. **c** Side view and **d** top view of the channels formed by the periplasmic domain of the form II structure of the CmeB trimer. The "extrusion" protomer generates an extrusion channel that is similar to those observed in the form I structure. The "binding" protomer constitutes a channel, which leads through the opening of the periplasmic cleft and is exposed to solvent. The "resting" protomer does not form any channel. These channels were calculated using the program CAVER (http://loschmidt.chemi.muni.cz/caver)

go through a sequence of conformational transitions, which lead to the extrusion of substrates through a particular protomer. The structures of individual protomers of AcrB and CmeB captured by crystallography may simply reflect the conformation of various transient states that these protomers may go through within the transport cycle. To this point, the conformations of the three protomers within the trimeric pump can be identical with three periplasmic clefts open or closed as shown in the case of the symmetric structures of AcrB[22] and CusA[19]. However, the structures of individual protomers of the trimeric pump can also be distinct from each other as indicated in the cases of the asymmetric AcrB structures[14, 17, 18], where the three protomers are in different transient states with two open and one closed periplasmic clefts within the trimer. For the asymmetric CmeB trimer, the conformations of the three protomers display in such a way that either only one out of the three periplasmic clefts is open or all of these clefts are closed.

**Transport dynamics of *C. jejuni* CmeB.** To elucidate if a CmeB protomer can export drugs individually within the trimer, we decided to directly observe the transport dynamics and conformational changes of the periplasmic domain movements using total internal reflection sm-FRET imaging. The CmeB protein has three cysteines (C453, C496, and C544) located at the transmembrane. We replaced these three cysteines by serines to create the cysteineless CmeB membrane protein. This cysteineless

CmeB pump is fully functional as suggested by in vivo susceptibility assay (Supplementary Notes, Supplementary Fig. 8 and Supplementary Table 4). On the basis of the crystal structures of CmeB, we then introduced a single cysteine mutation on the cysteineless protein at a position of high-solvent accessibility as well as low-sequence conservation. The resulting three cysteine residues within each CmeB trimer were derivatized with a mixture of maleimide-activated Alexa Fluor 546 (AF546) and Alexa Fluor 647 (AF647), which served as a molecular ruler for measuring the distance between two inter-subunit cysteines. Our goal was to measure the relative change in distance instead of absolute distance as the finite size, orientation, and length of fluorophores bring difficulty of measuring a precise distance through FRET[28, 29]. Thus, we mainly focused on the conformational movement of the CmeB transporter and made point mutation to observe the functional dynamics during drug export. As CmeB is a proton-motive-force (PMF)-dependent transporter, we reconstituted the purified and derivatized CmeB protein into liposomes, where we could generate the proton gradient required for substrate translocation. These proteoliposomes were immobilized on streptavidin-decorated surfaces for FRET signal recording (Fig. 3a).

We chose to mutate residue K843 of the cysteineless CmeB protein to a cysteine in order to anchor the dyes. This residue is located right outside subdomain PC2, facing the periplasm, and at a position where the inter-subunit distances are quite different

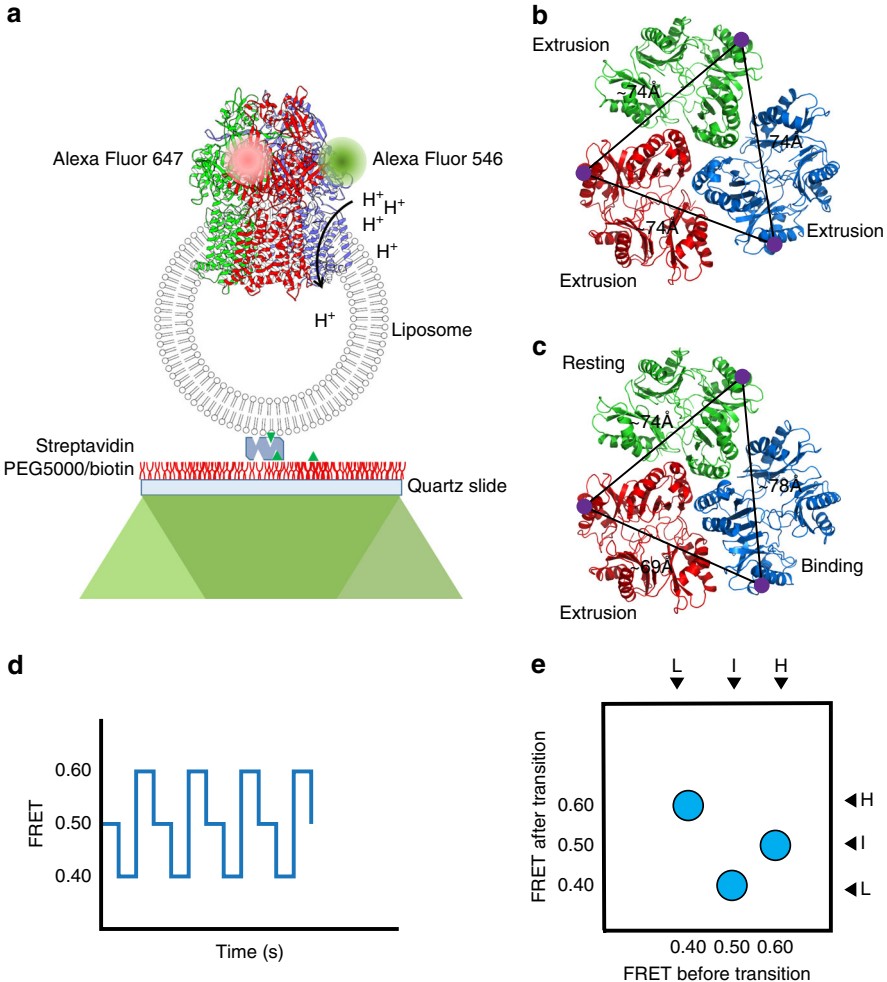

**Fig. 3** Schematic diagram for visualizing the dynamics of the trimeric CmeB pump incorporated in proteoliposomes using total internal reflection sm-FRET imaging. **a** The three subunits of CmeB embedded in the lipid vesicle are colored differently. The AF546 and AF647 dyes are in *green* and *magenta*, respectively. **b** Inter-subunit distances measured between the two K843 residues of the form I CmeB trimer. **c** Inter-subunit distances measured between the two K843 residues of the form II CmeB trimer. **d** The predicted FRET trajectory and **e** predicted density plot of the transport dynamics of the CmeB pump if this pump functions via the proposed rotating mechanism

between the conformations as suggested by the crystal structures (Fig. 3b, c). Both in vitro proton transport and in vivo susceptibility assays indicated that the K843C mutant is fully functional (Supplementary Notes, Supplementary Fig. 9 and Supplementary Table 4). We selected the CmeB trimers that only contained one donor (AF546) and one acceptor (AF647) dyes for FRET measurements. If CmeB functions by means of the proposed rotating mechanism[14, 17, 18], sequentially transitioning through three different states, then the distance between the two inter-subunit K843 residues should sequentially vary in a manner of ~74, ~79, and ~69 Å, respectively. This should lead to the observation of three distinct signals, which correspond to the intermediate-, low-, and high-FRET states, in the FRET trajectory and transition density plot (Fig. 3d, e). However, if CmeB employs a different mechanism to recognize and export substrates, then the characteristic of this transition density plot should be different.

We adjusted the extra- and intra-vesicular pHs of the CmeB K843C proteoliposomes to 6.5 and 7.5, respectively. We then performed sm-FRET experiments both in the absence and presence of 1 or 10 μM taurodeoxycholate (Tdc), which is the CmeB substrate. In the absence and presence of Tdc, the FRET state values are very similar. We also find that the frequency of

transitions is more or less the same with and without the substrate Tdc. The majority of the populations of apo-CmeB are largely in favor of the low FRET state. However, the addition of substrates seems to shift the state occupancies more favorable to the higher FRET states (Fig. 4). At least four distinct states can be observed, indicating that the trimeric pump is transitioning between various states. These four states are labeled as low (L), intermediate-1 ($I_1$), intermediate-2 ($I_2$), and high (H) FRET states, which correspond to ~0.20, ~0.35, ~0.45, and ~0.60 FRET efficiencies. Interestingly, our sm-FRET data do not seem to agree with the proposed rotating mechanism. On the basis of the traces and symmetrical nature of the density plots, it is more likely that the three protomers function independently of each other.

As a negative control, we selected to mutate residue K781 of the cysteineless CmeB pump to a cysteine. This residue is located at the top of the funnel region (subdomain DC). According to our crystal structures, it was predicted that this region should not have any significant motions. The inter-subunit distance between the two K781 residues is ~52 Å. This mutant behaves the same as the wild-type pump as suggested by the in vivo susceptibility assay (Supplementary Table 4). FRET efficiency distributions histogram yielded a single narrow peak centered at 0.85

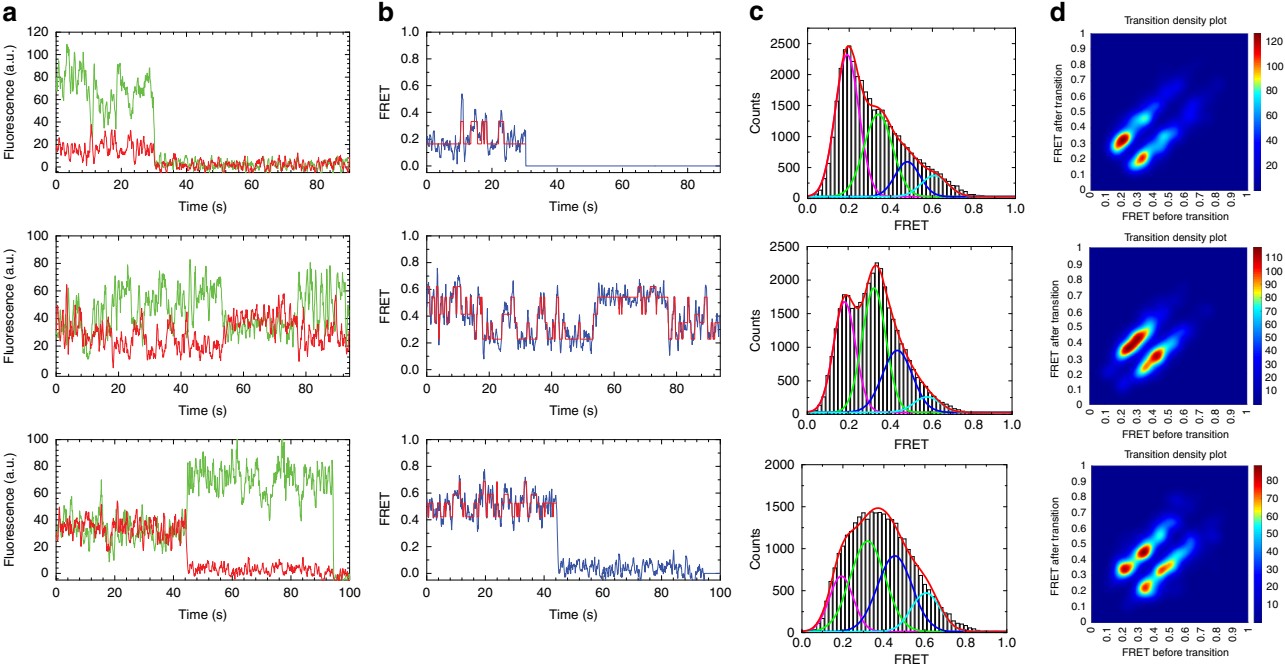

**Fig. 4** Dynamics in the apo and substrate bound CmeB pump. **a** Experimental trace of CmeB single-molecule dynamics with donor (*green*) and acceptor (*red*) fluorescence (in the absence of Tdc, *top panel*; in the presence of 1 μM Tdc, *middle panel*; in the presence of 10 μM Tdc, *bottom panel*). **b** FRET trajectories (*blue*) and idealization of these FRET efficiencies (*red*) (in the absence of Tdc, *top panel*; in the presence of 1 μM Tdc, *middle panel*; in the presence of 10 μM Tdc, *bottom panel*). **c** FRET efficiency population histograms of CmeB (in the absence of Tdc (*n* = 69 traces), *top panel*; in the presence of 1 μM Tdc (*n* = 77 traces), *middle panel*; in the presence of 10 μM Tdc (*n* = 73 traces), *bottom panel*). The histograms were fitted with Gaussian functions. **d** Transition density plots for the CmeB efflux pump (in the absence of Tdc, *top panel*; in the presence of 1 μM Tdc, *middle panel*; in the presence of 10 μM Tdc, *bottom panel*)

(Supplementary Fig. 10), indicating that this funnel region lacks a major conformational change throughout the efflux process. The data also demonstrated the robust performance of our FRET system.

Drug export by RND transporters depends upon the PMF[12]. In the transmembrane region of CmeB, the conserved charged residues D409, D410, and K935 form a salt-bridge triad, which most probably establishes the proton-relay network and relays proton translocation for energy coupling. A single point mutation on these corresponding residues in MexB[30], AcrB[31], and CusA[19, 27] has been found to impair the function of these pumps. To disrupt this proton-relay network in the CmeB pump, we replaced D409 by an alanine of the K843C pump to form a K843C-D409A double-point mutant. In vitro proton transport assay has confirmed that proton transport activity in the D409A mutant was totally abolished (Supplementary Fig. 9). This double-point mutant was then purified, derivatized with AF546 and AF647, and reconstituted into liposomes for FRET signal recording. Again, all of these sm-FRET experiments were done in the presence of a proton gradient with the extra- and intra-vesicular pHs at 6.5 and 7.5, respectively. As expected, this double mutant did not show much activity in terms of its dynamic movement. In the absence of Tdc, most of the population of this K843C-D409A mutant is found to cluster in the low FRET state regardless of the presence of ligand (Fig. 5). At the population level, a dominant single narrow peak at 0.2 FRET efficiency was observed (Fig. 5a). On the basis of the transition density plot (Fig. 5b), the frequency of transitions of the K843C-D409A mutant was reduced by at least seven times compared with that of the K843C transporter in the absence of Tdc. Only a small fraction of transitions between 0.2 and 0.35 FRET efficiency states was observed. However, they are almost undetectable in the FRET histograms. This low FRET state is also seen in the K843C mutant. The FRET data indicate

that the distance between the donor and acceptor dyes at this low FRET state is ~85 Å, corresponding to the 0.2 FRET efficiency. This distance is significantly longer than the distances observed in our crystal structures. However, this distance is similar to that depicted in the crystal structure of the "resting" state of apo-CusA[19, 20]. In this state, the three periplasmic clefts of the CusA heavy-metal efflux pump are closed within the trimer. The data suggest that the CmeB protomers may prefer the "resting" conformation, in the absence of ΔpH, and the process of transitioning from the "resting" to "binding" states may be energy dependent.

Hidden Markov modeling[32] was then used to quantify the transition rates of these various FRET states. In the absence of Tdc, the transition density plot of the K843C mutant indicates that the predominant FRET transitions are L → I$_1$ and I$_1$ → L, which correspond to the reversible transitions between the "resting-resting" and "resting-binding" subunits. A histogram derived from a population of dwell times was fitted with a single exponential decay, resulting in $k_{L→I1}$ = 1.64 s$^{-1}$ (0.61 s) and $k_{I1→L}$ = 3.22 s$^{-1}$ (0.31 s) transition rates for the processes "resting-resting" → "resting-binding" and "resting-binding" → "resting-resting", respectively (Table 2 and Supplementary Fig. 11). The data suggest that the kinetics of CmeB are quite simple and can be described with a single rate constant.

In the presence of 1 μM Tdc, there is a significant decrease in the reverse transition I$_1$ → L. The rate for this reverse transition is $k_{I1→L}$ = 1.82 s$^{-1}$ (0.55 s), which is almost two times slower than the same process without the substrate. In addition, there is a substantial increase in the forward transition I$_1$ → I$_2$, suggesting that the K843C mutant may prefer to advance the transport cycle by shifting the "resting-binding" to "resting-extrusion" or "binding-binding" states. Apparently, this process is reversible as indicated by the observation of the reverse transition I$_2$ → I$_1$.

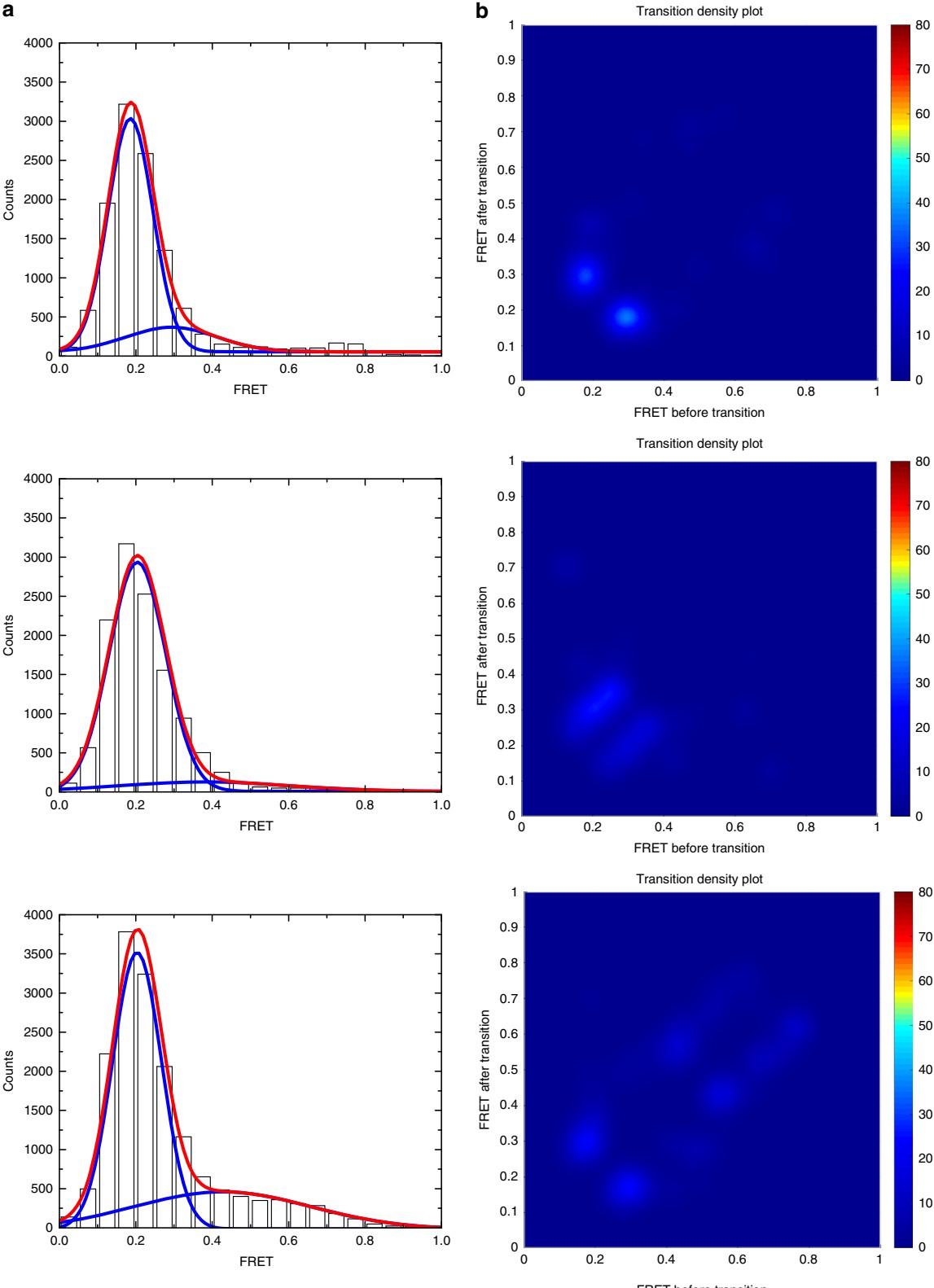

**Fig. 5** Dynamics in the apo and substrate bound of the proton-relay mutant of CmeB. **a** FRET efficiency population histograms of CmeB (in the absence of Tdc ($n = 40$ traces), *top panel*; in the presence of 1 μM Tdc ($n = 37$ traces), *middle panel*; in the presence of 10 μM Tdc ($n = 47$ traces), *bottom panel*). **b** Transition density plots for the CmeB efflux pump (in the absence of Tdc, *top panel*; in the presence of 1 μM Tdc, *middle panel*; in the presence of 10 μM Tdc, *bottom panel*)

**Table 2 Dwell times for CmeB transitions**

| CmeB | [Tdc] (µM) | FRET transition dwell time (s) | | | | | |
|---|---|---|---|---|---|---|---|
| | | $L \rightarrow I_1$ | $I_1 \rightarrow L$ | $I_1 \rightarrow I_2$ | $I_2 \rightarrow I_1$ | $I_2 \rightarrow H$ | $H \rightarrow I_2$ |
| K843C | 0 | 0.61 ± 0.06 | 0.31 ± 0.04 | 0.54 ± 0.04 | 0.26 ± 0.07 | | |
| | 1 | 0.52 ± 0.04 | 0.55 ± 0.05 | 0.62 ± 0.03 | 0.29 ± 0.03 | | |
| | 10 | 0.47 ± 0.03 | 0.65 ± 0.02 | 0.50 ± 0.03 | 0.49 ± 0.07 | 0.63 ± 0.07 | 0.35 ± 0.03 |
| K843C−D409A | 0 | 1.85 ± 0.13 | 0.35 ± 0.01 | | | | |
| | 1 | 1.68 ± 0.34 | 0.31 ± 0.05 | | | | |
| | 10 | 1.47 ± 0.39 | 0.39 ± 0.12 | | | | |

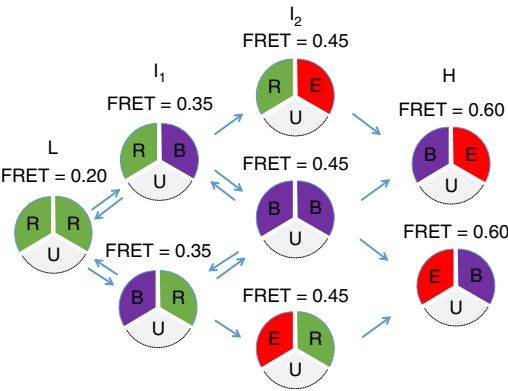

**Fig. 6** Proposed model of drug efflux mechanism. On the basis of our experimental data, there are at least four distinct states found in the trimeric CmeB multidrug efflux pump. These four states are labeled as low (*L*), intermediate-1 (*I₁*), intermediate-2 (*I₂*), and high (*H*) FRET states, which correspond to ~0.20, ~0.35, ~0.45, and ~0.60 FRET efficiencies. During drug export, each protomer of CmeB autonomously undergoes a sequence of conformational transitions through a particular protomer. This schematic diagram indicates that each protomer within the CmeB trimer can independently go through a sequence of conformational transitions, which lead to the extrusion of substrate (*R* resting, *B* binding, *E* extrusion, *U* unlabeled)

The forward and reverse transition rates were calculated to be $k_{I1 \rightarrow I2} = 1.61\ \text{s}^{-1}$ (0.62 s) and $k_{I2 \rightarrow I1} = 3.45\ \text{s}^{-1}$ (0.29 s), respectively (Table 2 and Supplementary Fig. 12).

As the Tdc concentration increased to 10 µM, the transition between $I_2$ and $H$ was observed, suggesting that the transporter continues to move forward the transport cycle by switching from the "resting-extrusion" to the "binding-extrusion" forms. This process is also reversible and the transition rates for these forward and reverse processes are $k_{I2 \rightarrow H} = 1.59\ \text{s}^{-1}$ (0.63 s) and $k_{H \rightarrow I2} = 2.86\ \text{s}^{-1}$ (0.35 s) (Table 2 and Supplementary Fig. 13).

For the K843C-D409A mutant pump, the density plots are much simpler and their characteristics are more or less the same regardless of the presence of Tdc. Both in the absence and presence of substrate, the only observed transitions are the reversible conformational change between the "resting-resting" and "resting-binding" forms, as indicated by the $L \rightarrow I_1$ and $I_1 \rightarrow L$ transitions. However, the rates of forward transitions are much slower than those for the K843C mutant. Specifically, in the absence of Tdc, the rates for the forward and reverse transitions in the K843C-D409A pump are $k_{L \rightarrow I1} = 0.54\ \text{s}^{-1}$ (1.85 s) and $k_{I1 \rightarrow L} = 2.86\ \text{s}^{-1}$ (0.35 s) (Table 2 and Supplementary Fig. 14), suggesting that the transition process "resting-resting" → "resting-binding" may be energy dependent and needs to couple with ΔpH. All of these observed transition rates are listed in Table 2.

## Discussion

We have defined the crystal structures of CmeB and directly observed the transport dynamics of this membrane protein reconstituted in proteoliposomes at the single-molecule level. These data lead us to propose a simple model for the CmeB transport mechanism (Fig. 6), in which the protomers export substrates independently of each other within the trimer. In the absence of ΔpH, the CmeB protomers may prefer the "resting" conformation, which is evidenced through our FRET data that the inter-subunit distance of the K843C-D409A double mutant measured between the two K843 residues is relatively long in comparison with that of the K843C mutant. It was found that there are relatively very little motions going on in this double mutant, suggesting that a transition from the "resting" to "binding" states may need to couple with ΔpH. In the presence of ΔpH, it appears that there are a few more observed transitions, suggesting that the pump can easily continue to advance the transport cycle by coupling with the proton-relay network. Our data indicate that the CmeB protomers can independently progress to "binding" and then "extrusion" conformations. The populations of these two states can be greatly enhanced by the addition of the Tdc ligand. Our data allow us to uncover the mechanism of drug export, where the three CmeB subunits undergo conformational changes independently of each other.

## Methods

**Expression and purification of CmeB.** The CmeB multidrug efflux pump, which contains a 6xHis tag at the N terminus was overexpressed in *E. coli* BL21(DE3) Δ*acrB*/pET15bΩ*cmeB* cells. These cells harbor a deletion in the chromosomal *acrB* gene and possess the pET15bΩ*cmeB* vector. Cells were grown in 12 litres of Luria-Bertani (LB) broth supplemented with 100 µg ml⁻¹ ampicillin at 37 °C. When the OD₆₀₀ reached 0.5, cells were treated with 1 mM IPTG to induce *cmeB* expression and harvested within 3 h. The collected cells were resuspended in buffer containing 100 mM sodium phosphate (pH 7.2), 10% glycerol, 1 mM ethylenediaminetetraacetic acid (EDTA), and 1 mM phenylmethanesulfonyl fluoride (PMSF). The suspension was then disrupted using a French pressure cell. The membrane fraction was collected and washed twice with buffer containing 20 mM sodium phosphate (pH 7.2), 2 M KCl, 10% glycerol, 1 mM EDTA, and 1 mM PMSF, and once with 20 mM HEPES-NaOH buffer (pH 7.5) containing 1 mM PMSF. The CmeB membrane protein was then dissolved in 1% (w:v) 6-cyclohexyl-1-hexyl-β-D-maltoside (Cymal-6). Insoluble material was taken away by ultracentrifugation at 100,000×*g*. The extracted CmeB membrane protein was purified with a Ni²⁺-affinity column. The purity of the protein (>95%) was judged using 10% sodium dodecyl sulfate-polyacrylamide gel electrophoresis (SDS-PAGE) stained with Coomassie brilliant blue. The purified protein was then dialyzed and concentrated to 20 mg ml⁻¹ in a buffer containing 20 mM Na-HEPES (pH 7.5) and 0.05% Cymal-6.

**Crystallization of CmeB.** The 6xHis CmeB crystals were grown at room temperature using sitting-drop vapor diffusion with the following procedures. A 2 µl protein solution containing 20 mg ml⁻¹ CmeB protein in 20 mM Na-HEPES (pH 7.5) and 0.05% (w:v) Cymal-6 was mixed with a 2 µl of reservoir solution containing 4% PEG 8000, 0.1 M Na-MES (pH 6.5) and 0.1 M MgSO₄. The resultant mixture was equilibrated against 500 µl of the reservoir solution. Crystals of CmeB grew to a full size in the drops within a month. Typically, the dimensions of the crystals were 0.2 mm × 0.2 mm × 0.2 mm. Cryoprotection was achieved by raising the glycerol concentration stepwise to 25% with a 5% increment in each step.

**X-ray structural determination and refinement**. X-ray diffraction data were collected at 100 K at beamline 24ID-C located at the Advanced Photon Source with a Platus 6M detector. The data were processed and scaled using DENZO and SCALEPACK[33], respectively.

The form I crystals of CmeB belong to space group C2 (Table 1). The structure of CmeB was phased using molecular replacement. On the basis of the structure of the O protomer of AcrB (PDB ID: 4dx5)[34], a model of CmeB generated by the FFAS03 server was used as a search model. The initial $R_{work}$ and $R_{free}$ were 34.9 and 42.0%, respectively. After tracing the initial model manually using Coot[35], the model was refined to a resolution of 3.15 Å using PHENIX[36]. Iterations of refinement were done using PHENIX[36] and CNS[37]. Model building was carried out using Coot[35]. Except for the last seven residues at the C-terminal end, all of the amino acids within the CmeB molecule were assigned. The final CmeB model consists of 3,105 residues in the asymmetric unit with excellent geometrical characteristics (Table 1).

The form II crystals took the space group P1 (Table 1). This structure was determined using molecular replacement. The structure of form I was employed as a search model. After tracing the initial model using Coot[35], the model was refined to a resolution of 3.55 Å. The remaining procedures for model building and structural refinement were identical to those for the form I structure.

**Protein reconstitution into liposomes**. Single cysteine mutations were introduced to produce a cysteineless CmeB variant, in which the three natural cysteine residues located in the transmembrane region have been replaced with serines, using site-directed mutagenesis. Constructs were verified by DNA sequencing and transformed into E. coli BL21(DE3)ΔacrB cells. Proteins were expressed as N-terminal 6xHis fusions and purified as described above. The purified proteins were then dialyzed, concentrated to 20 μM in a buffer containing 20 mM Na-HEPES (pH 7.5) and 0.03% DDM, and labeled with a mixture of maleimide-activated AF546 and AF647 at final concentrations of 20 and 100 μM, respectively. Labeled proteins were quenched with 10 mM 2-mercaptoethanol and subsequently purified away from excess reagents using size exclusion chromatography.

The labeled CmeB variants were reconstituted into liposomes made of 74.5% E. coli total lipid, 24.5% egg-yolk phosphatidylcholine, and 1% 1,2-dioleoyl-sn-glycero-3-phosphoethanolamine-N-(cap biotinyl) (biotin-DOPE) (Avanti Polar Lipids, Alabaster, AL) in a buffer containing 20 mM Na-HEPES (pH 7.5). Each CmeB variant was mixed with unilamellar liposomes in 20 mM Na-HEPES (pH 7.5) and 0.2% n-dodecyl-β-D-maltoside (DDM) at a protein-to-lipid ratio of 1:10,000 (w:w) to ensure a high probability of having only one single CmeB trimer per liposome. The liposome-protein mixture was incubated for 1 h at room temperature under gentle agitation. Subsequently, this mixture was diluted stepwise three times within 45 min. The final concentration of DDM should be below the critical micelle concentration of ~0.008%. Detergents were then removed by SM2 Biobeads (Bio-Rad) and PD-10 desalting column (GE Healthcare Bio-Sciences, Pittsburgh, PA).

FRET between neighboring proteins on the same liposome could be a problem because of the inter-molecular FRET. Wang et al.[28] mixed unlabeled and labeled protein for reconstitution into liposomes. This is indeed a great strategy to avoid this problem. However, it is not easy to maintain proton gradient in heterogenic proteoliposomes due to various quantity of transporters per proteoliopsome. Therefore, we decided to use low protein-to-lipid ratio method[38] to ensure a small probability of 0.03% of trapping multiple CmeB trimers in a single 100-nm liposome.

**Single-molecule FRET experiments**. All sm-FRET experiments were performed using a home-built prism-based total internal reflection fluorescence microscope constructed around an inverted microscope body (Olympus IX71). The samples were illuminated with a 532 nm solid-state laser to excite the AF546 donor dye. A 635 nm helium-neon laser was used to ensure the presence of the AF647 acceptor dye. The AF546 and AF647 fluorescence signals were collected using a water-immersion lens (Olympus UPLSAPO 60XW) and separated using a dichroic mirror (Chroma T6601pxr). Imaging data were acquired using smCamera acquisition software and an electron-multiplying charged-coupled device camera (Andor Technology iXon3 DU879E).

Quartz slide (Chemglass Life Sciences, Vineland, NJ) and micro cover glass (VWR Life Sciences, Radnor, PA) were extensively cleaned and functionalized by coating the surface with methoxy-PEG-5000 and Biotin-PEG-5000 (100:1) (Laysan Bio, Arab, AL). These slides and cover glasses were assembled to form a flow cell device. This flow cell was then incubated with a solution containing 20 mM Na-HEPES (pH 7.5) and 100 μg ml⁻¹ streptavidin from Streptomyces avidinii (Sigma-Aldrich, St. Louis, MO) for 5 min. Unbound streptavidin was subsequently washed out with 20 mM Na-HEPES (pH 7.5). Next, a suspension of proteoliposomes (100 μg ml⁻¹ lipid concentration) extruded through a 100-nm pore-size polycarbonate filter (GE Healthcare Bio-Sciences, Pittsburgh, PA) was flushed in, followed by a 4-min incubation to allow the liposomes to adhere to the surface. Unbound proteoliposomes were washed away with buffer containing 20 mM Na-HEPES (pH 7.5).

All imaging experiments were performed in solution containing 20 mM HEPES (pH 6.5), 2 mM cyclooctatetraene (Sigma-Aldrich, St. Louis, MO) and 5 mM

β-mercaptoethanol (Sigma-Aldrich, St Louis, MO). Cyclooctatetraene is an efficient triplet state quencher that reduces the lifetime of dark states[39]. According to our ITC data, the presence of this compound does not interfere with the binding affinity of Tdc by the CmeB pump (Supplementary Notes, Supplementary Fig. 15, and Supplementary Table 5). To avoid unwanted pH drop causing by common enzymatic oxygen scavengers, such as glucose oxidase and catalase and protocatechuate dioxygenase, during data collection, we used a pH stable pyranose oxidase and catalase[40] in all of our experiments. Thus, the solution was supplemented with a pH stable enzymatic oxygen scavenger system comprising 3 units ml⁻¹ pyranose oxidase, 8 units ml⁻¹ catalase, and 0.8% glucose (Sigma-Aldrich, St Louis, MO). All data were collected at an imaging rate of 10 s⁻¹ (100 ms integration time). At the beginning of each experiment, a 10 s (100 frames) movie was recorded with an alternation of 532- and 637-nm excitation. This alternating laser excitation (ALEX) scheme[41] allowed us to separate the fluorescence contributions of the green and red dyes, thus permitting us to identify the CmeB trimers that contain only one green and one red dyes.

**Single-molecule FRET data analysis**. The movies acquired in single-molecule imaging were processed with smCamera program (https://cplc.illinois.edu/software/) to identify and extract donor and acceptor fluorescence intensity profiles of individual molecules. Traces extracted from the movies were interactively selected with the following criteria: (i) only a single AF546 and a single AF647 dyes in a proteoliposome determined by the ALEX scheme[41] was used; (ii) no more than one bleaching step for the donor and acceptor fluorophores was allowed; (iii) only those data indicate a clear anti-correlated pattern between the donor and acceptor fluorescence intensities were used; and (iv) vesicles that contain a constant total fluorescence intensity from the donor and acceptor before photobleaching were selected. Approximately 10% of the molecules matched these criteria. The FRET trajectories were calculated from the acquired intensities, $I_{AF546}$ and $I_{AF647}$, using the formula FRET = $I_{AF647}/(I_{AF546} + I_{AF647})$. Individual single-molecule traces were analyzed using HaMMy[32] to generate idealized traces. Transition density plots were generated using the MATLAB code[42], which was kindly provided by Professor Jong-Bong Lee from POSTECH, Korea. The histograms were calculated using the Origin Pro software (OriginLab, Northampton, MA). FRET histograms were compiled from proteoliposomes that only contain one donor and one acceptor dyes[43]. Maximum-likelihood estimate (MLE)[28] was used for peak fitting (Figs. 4 and 5). The MLE code was obtained from Professor Yong Wang at the University of Arkansas. This method does not require binning the data before fitting. After MLE fitting, the data are binned based on the fitting result. For fitting multiple Gaussians peaks in the K843C data, the peak centers were determined by HHM using the program Origin Pro (OriginLab, Northampton, MA).

**Construction of plasmid for chromosomal complementation**. Plasmids for chromosomal complementation were constructed by PCR amplifying the intact cmeABC operon from genomic DNA of C. jejuni 81-176 with primers 5′-GTTGG ATCACCTCCTT TCTAGATTTATTTAGAGAATAGAACAAACTATTC-3′ and 5′-GATAAGCGCGCTGCCTATCAATTGTCTAGAAAAATAATTTTATTAACC AAAATTAAG-3′. The corresponding PCR product was extracted from the agarose gel. The pRRK vector was digested with XbaI (NEB, MA) and gel purified. The purified PCR product of cmeABC operon was inserted into linearized pRRK vector using the SLiCE[44] method, resulting in the creation of pRRKcmeABC. The recombinant plasmid was transformed into DH10b cells and selected on LB agar plates containing 30 μg ml⁻¹ kanamycin. The presence of the correct cmeABC and cmeR sequence in the plasmid construct was verified by DNA sequencing. Site-directed point mutations on residues were performed by PCR to generate point mutants, including D409A, D410A, cysteineless (C544S, C496S, and C453S), cysteineless K843C, and cysteineless K781C. All oligonucleotides were purchased from Integrated DNA Technologies, Inc. (Coralville, IA, USA).

**Complementation of the ΔcmeABC::Cmr mutant**. The ΔcmeABC::Cm^r mutant was complemented by inserting a wild-type copy of cmeABC and cmeR between the 16S and 23S rRNAs as described by Muraoka and Zhang[45]. Briefly, the pRRKcmeABCR was used as the suicide vector to insert the cmeABC and cmeR genes into the chromosome of the ΔcmeABC mutant. The complemented strains were selected on MH agar plate containing 30 μg ml⁻¹ of kanamycin to screen for the transformants. The positive transformants were further confirmed by PCR.

**SDS-PAGE and immunoblotting**. To prepare whole-cell lysates, Campylobacter strains were grown in MH broth to late logarithmic phase ($3 \times 10^9$ cells ml⁻¹), harvested by centrifugation, and solubilized by boiling for 5 min in SDS-PAGE sample buffer. Ten-microgram membrane fractions or ~$10^8$ whole cells were loaded in each lane and separated by SDS-PAGE with a 10% (wt:vol) polyacrylamide separating gel. After SDS-PAGE, the gels were then electro-phoretically transferred to nitrocellulose membranes (Bio-Rad) at 60 V for 1 h at 4 ° C. The membranes were incubated with blocking buffer (5% Nestle skim milk powder in PBS) for 16 h at 4 °C prior to incubation with primary antibodies (rabbit anti-CmeB and anti-CmeC serum; 1:1,000 dilution in the blocking buffer). After incubation at 25 °C for 1 h, the blots were washed three times with PBS containing

0.05% Tween 20 and subsequently incubated with secondary antibodies (1:1,000 dilution of goat anti-rabbit immunoglobulin G-horseradish peroxidase; Kirkegaard & Perry) at 25 °C for 1 h. After washing, the blots were developed with the four CN Membrane Peroxidase Substrate System (Kirkegaard & Perry). Prestained molecular mass markers (Bio-Rad, Hercules, CA) were co-electrophoresed and blotted to allow estimation of the sizes of the proteins.

**Drug susceptibility assay**. The MICs of sodium taurocholate, sodium taurodeoxycholate, and rifampin for *C. jejuni* 81-176 and its *cmeABC* mutant constructs were determined using a microtiter broth dilution method as described previously[9]. The compounds utilized in these assays were purchased from Sigma-Aldrich (St Louis, MO).

**Isothermal titration calorimetry**. We used isothermal titration calorimetry (ITC) to examine the binding of taurodeoxycholate to the purified CmeB transporter. Measurements were performed on a VP-Microcalorimeter (MicroCal, Northampton, MA) at room temperature. The protein was thoroughly dialyzed against buffer containing 20 mM Na-HEPES pH 7.5 and 0.03% n-dodecyl-β-D-maltoside (DDM). Its concentration was determined using the Bradford assay and then adjusted to a final monomeric concentration of 20 µM before titration. Ligand solution consisting of 0.25 mM taurodeoxycholate in 20 mM Na-HEPES pH 7.5 and 0.03% DDM was employed as the titrant. The protein and ligand samples were degassed before loading into the cell and syringe. ITC experiments were performed with the protein solution (1.5 ml) in the cell and the ligand solution as the injectant. Ten microliter injections of the ligand solution were used for a set of data collection.

Injections occurred at intervals of 300 s. The duration time of each injection was 20 s. Heat transfer (µcal s$^{-1}$) was recorded as a function of elapsed time (s). The mean enthalpies measured from injection of the ligand in the buffer were subtracted from the raw titration data before data analysis using the ORIGIN software (MicroCal, Northampton, MA). Titration curves were fitted with a nonlinear least squares function. The nonlinear regression fitting to the binding isotherm allowed us to quantify the equilibrium binding constant ($K_A = K_D^{-1}$) and enthalpy of binding ($\Delta H$). On the basis of the values of $K_A$, the change in free energy ($\Delta G$) and entropy ($\Delta S$) were obtained using the equation: $\Delta G = -RT \ln K_A = \Delta H - T\Delta S$, where $T$ is 273 K and $R$ is 1.9872 cal K$^{-1}$ mol$^{-1}$. Calorimetry trials were also carried out in the absence of the CmeB protein in the same experimental conditions. No change in heat was observed in the injections throughout the experiment.

**In vitro proton translocation assay**. The wild-type, K843C and D409A CmeB transporters were reconstituted into liposomes made of 75% *E. coli* total lipid, 25% egg-yolk phosphatidylcholine (Avanti Polar Lipids, Alabaster, AL) and 2 mM 8-hydroxypyrene-1,3,6-trisulfonic acid trisodium (Sigma-Aldrich, St Louis, MO) in a buffer containing 20 mM Na-HEPES (pH 7.5). The CmeB protein (wild-type, K843C or D409A) was mixed with unilamellar liposomes in 20 mM Na-HEPES (pH 7.5) and 0.2% DDM at a protein-to-lipid molar ratio of 1:20 (w:w). The liposome-protein mixture was then incubated for 1 h at room temperature under gentle agitation. Subsequently, this mixture was diluted stepwise three times within 45 min. The final concentration of DDM should be below the critical micelle concentration of ~0.008%. The detergent molecules were then removed using SM2 Biobeads (Bio-Rad) and PD-10 desalting column (GE Healthcare Bio-Sciences, Pittsburgh, PA).

Before measurement, the proteoliposomes were extruded through a 100-nm pore-size filter and purified using a PD-10 column to remove untrapped dyes. All measurements were performed at 25 °C using a PerkinElmer LS55 spectrofluorometer equipped with a Hamamatsu R928 photomultiplier. The excitation and emission wavelengths were 460 and 510 nm, respectively. To reduce the external pH, an equal volume of 20 mM Na-HEPES (pH 6.5) buffer was added to the solution. To calculate the relative fluorescence change, the protonophore carbonyl cyanide *m*-chlorophenylhydrazone (CCCP) was then added at 50 µM to disrupt proton gradient. $F_{min}$ represents the fluorescence intensity after adding CCCP, while $F_{max}$ corresponds to the fluorescence intensity at time zero.

**Data availability**. Atomic coordinates and structure factors for the structures of CmeB have been deposited at the RCSB Protein Data Bank with accession codes 5T0O (form I) and 5LQ3 (form II).

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

## Acknowledgements

This work was supported by NIH Grants R01AI114629 (E.W.Y.), R01GM051290 (Y.K.S.), 5U54GM087519 (Y.K.S.), and R01AI118283 (Q.Z.). This work is based upon research conducted at the Northeastern Collaborative Access Team beamlines of the Advanced Photon Source, supported by an award GM103403 from the National Institutes of General Medical Sciences. Use of the Advanced Photon Source is supported by the U.S. Department of Energy, Office of Basic Energy Sciences, under Contract No. DE-AC02-06CH11357.

## Author contributions

C.-C.S and E.W.Y. designed crystallography experiments. C.-C.S., N.K., J.R.B., and H.-T. L. cloned, expressed, and purified CmeB. C.-C.S., J.R.B. and H.-T.L. crystallized form I CmeB. N.K. and A.R. crystallized form II CmeB. C.-C.S., N.K., A.R., J.R.B., H.-T.L., T.-H.C., and K.R.R. collected X-ray diffraction data. C.-C.S. performed model building and refinement of the CmeB structures. C.-C.S., L.Y., and Y.-K.S. designed single-molecule FRET experiments. C.-C.S. and L.Y. performed single-molecule FRET experiments. C.-C.S., L.D., and Q.Z. designed drug susceptibility experiments. L.D. performed drug susceptibility experiments. C.-C.S. and E.W.Y. wrote the paper. C.-C.S., J.A.D., and E.W.Y. proofread the paper. E.W.Y. supervised the research.

## Additional information

**Competing interests:** The authors declare no competing financial interests.

