## [Peer Review File · Nature Communications]

Reviewers' comments:

Reviewer #1 (Remarks to the Author):

The authors present two new structures of an RND efflux pump from *Campylobacter jejuni* and present single molecule FRET analysis of dynamics within the reconstituted CmeB. They postulate an interesting hypothesis of individual catalysis of the protomers within the trimer and challenge the other hypothesis of interdependent protomer cycling between access, binding, and extrusion states. The authors lab has already in the past crystallized different RND transporters in different states and are well-known within the field. I welcome the stochastic concept of protomer conformational change independent of the conformational change of the other protomers within the trimer. Having said this, I also think that much more experiments should be conducted to truly state that CmeB transports drugs in this way and I do not think that at the current state and content the manuscript should be published in Nature Communications.

Structure:

The authors solved the structure of CmeB at moderate resolution from crystals grown in two different space groups. In one trimeric structure, the protomers are in the extrusion state, and in the other structure two protomers are in the extrusion state and one protomer in the binding state.

- From Figure 1 and 2, it is very difficult to relate to the binding and extrusion states of the prototype AcrB, from where the designations come from. A defined table of the rmsd's of the PN1/PC2 subdomains, and PN2/PC1 subdomains, as well as the two parallel halves of the transmembrane domains must be given. Additionally, a figure with the differences between AcrB and CmeB differences in the regions with largest deviation should be given. This is also useful for the differences between the extrusion states in trimer form I and trimer form II (especially since a "new state" is mentioned).

- The extrusion states in form I and form II appear different. Moreover, the calculated channels between the extrusion protomers in form I and form II are different. Are the channels at its narrowest points large enough for substrates to passage? In form I, the tilted helix from the PN1 subdomain appears to block the channel more than the extrusion state in form II, where the neighboring protomer is in the binding state and does not have a tilted helix (or does it?). Why does the green protomer ("new state") in fig. 2d not have an extrusion channel? What is different compared to the other extrusion state protomers?

- Do all the extrusion protomers have the same transmembrane domain helix and side chain conformations (especially the Asp's and Lys at helix 4 and 10)?

- Which pdb of reference 14 has been used for MR? Why use a 2.8 Å AcrB model with modest geometry when a 1.9 Å structure (with better geometry) is available?

- The reported Rfree is very low considering the resolution given (especially for the P1 crystal). Very often the low Rfree still comes from the search model which has not been "shaken" in the refinement. Moreover, also the Rwork values are extremely low given the resolutions. Was the initial Rwork/Rfree after MR and the first cycle of refinement in the expected range (35-40%) and has this been consecutively lowered by model building and refinement rounds?

- Could the entire CmeB molecule be assigned (in both crystal forms)?

- A picture (close-up) of the distal binding pocket of CmeB should be provided (how different is it from AcrB). Could the binding protomer in principle bind drug and have co-crystallization trials been tried?

- What is the reason that form II is in the extrusion-extrusion-binding conformation? I guess that crystal contacts made with the binding protomer confer this asymmetry?

- Lines 136-140: Although I find the independent cycling of the protomers an interesting hypothesis, I don't think MdtBC is a good example for independent cycling. In fact, it is an example for dependent cycling, since two of the protomers are doing the energy transduction (and can't bind drugs), whereas the other is involved in drug binding.....(and can't transduce)(Kim et al., 2012). Unless the protomers are interacting in a concerted way, one cannot explain why the protomers incapable of energy transduction do expel the drugs (which is energy-dependent).

Functional studies:

- In line 154 it is stated: "To elucidate if a CmeB protomer can export drugs individually within the trimer,....", however, despite FRET analysis will indicate dynamics of the transporter, it will not tell whether the reconstituted CmeB is actually transporting the drug (in this case Tde).

- The CmeB single Cys variants were made in a Cys-less background, where the cognate Cys residues have been replaced by Ser. Is this Cys-less variant (and the single substitution variant K834C) still active (MIC, fluorescent substrate extrusion e.g.)? This must be tested.

- The pmf consist of two components (ΔpH and $\Delta \psi$). Both components should be tested alone and in combination in the assay and their effect on the dwelling times.

- I think it is important to test whether protons are transported across the membrane in the assay used. The use of pyranine dye might be helpful for this. Unless the proteoliposomes are leaky for protons (or maybe anions going inside the proteoliposomes, I would argue that the transported protons would create within seconds an opposite $\Delta \psi$ and stall further transport.

- How stable is the ΔpH applied (should be tested)? Since measurements are done for up to 100s, is the ΔpH still present?

- Without indication that protons are transported from outside to inside into the vesicles, data on conformational dynamics could also be interpreted as substrate binding, proton binding and their release to the same side where these solutes came from (the interproteoliposomal space), but no actual transport occurs (and hence no complete cycle of binding-extrusion-binding, but rather binding-release-binding). So the statement in lines 191-192 is almost correct: "Based on the traces and symmetrical nature of the density plots, it is more likely that the three protomers function independently of each other". However, the question is: do they transport the drug (and protons)?

- How stable are the proteoliposomes in the presence of up to 10 μM Taurodeoxycholate? Is there no interference with the integrity of the proteoliposomes? (Is the ΔpH affected?)

- I think another FRET pair should be tested as negative control, i.e. a pair which does not change its distance upon changing into the different conformational states (based on the structure).

- The frequency of transitions is reduced 7 times for the D409A variant. This appears not so much for a transporter which is incapable of transporting drugs and H^+ (again, this has to be shown in a phenotypical assay for CmeB as well, otherwise it is just an assumption). I agree that this can be interpreted that certain states still can be probed by the transporter, but why have not only less states, but also slower transitions? Do protons still have binding capacity in the transporter? Would the transitions be completely vanished if a double variant (also replacing the second Asp) would be tested?

- I have a conceptual question on the description of the resting state. The resting state has its cleft close and theoretically no access path for drugs. If I understand it correctly, the opening and conversion into a binding state would be pmf dependent. Under physiological conditions, when does this occur? Cells always have pmf (unless they are dead), so why have a resting state? On the other hand, if drug would be a trigger (which seems at least intuitively logical), where is the drug binding site in the resting state with the closed cleft?

- Related to this, line 221 states that one of the transitions in the absence of drug is considered "resting-resting". Nevertheless, in the proteoliposomal FRET experiment, a Δ pH is present. If the resting state is only in the absence of pmf, how is this possible (or I misunderstood the definition of "pmf-dependent").

- In lines 248-250, the statement couples the presence of pmf and substrate to be triggers for resting to binding conversion. If both are necessary, the question remains when the cell lacks the pmf in any stage of its life?

- A technical question: Why is cyclooctatetraene (2 mM) added to the imaging experiments. Could it be a substrate of the pump as well and has this been checked?

- Lines 400-406, what was the % of the total proteoliposomes (traces) which did match with the four criteria?

Reviewer #2 (Remarks to the Author):

The manuscript by Su et al. describes a model for the transport mechanism of the subunits of the CmeB trimer by proposing that the subunits function independently rather than in a cooperative manner. Given that CmeB is an RND-type multi-substrate efflux transporter, which contributes to bacterial resistance to a range of structurally unrelated antibiotic compounds, understanding structural dynamics of this transporter is important since it may contribute towards development of novel compounds against multidrug resistant *C. jejuni*. The manuscript documents the use of a number of complementary experimental techniques including X-ray crystallography, single-molecule FRET (smFRET) spectroscopy and Hidden Markov modeling (HMM) to support the proposed model. While the model is interesting I have several comments that are detailed below:

1. On p. 7, lines 128-135 the authors write: "Our structures of CmeB ... that individual protomers of these trimeric RND pumps could bind and export substrates independently instead of operating in a synchronized fashion." If the individual protomers can function independently why do they have to assemble into trimers? Why not monomers or pentamers, for example? Please comment.
2. p. 8, line 166: Given that the sm-FRET data do not agree with the proposed rotating mechanism what is the evidence that replacing K843 to a cysteine for the attachment of fluorophores does not alter the function of CmeB?
3. Transition density plots in Fig 4c and Fig. 5a should be fitted with Gaussians. The number of photobleaching steps, which tells the number of fluorophores attached to a trimer, does not seem to correspond to the number of Gaussians that would be expected in Fig 4c (e.g. top) and 5a (e.g. bottom). The bin size was set at 0.02 (p. 19, line 412). Given that the peaks of the distributions can be affected with different bin sizes and the selection of the bin size is subjective, the proper way for selection of the bin size is by using maximum-likelihood estimation for peak fitting.
4. FRET between neighboring CmeB trimers on the same liposome could be a problem because of inter-molecular FRET. What percentage of labeled vs. unlabeled trimers was used for reconstitution in liposomes? Justify.
5. Which precaution was taken to avoid the effect of the finite size and orientation of fluorescent probes bringing additional difficulty to converting FRET measurements to the estimation of distances?

6. p. 16, line 336: "Protein reconstitution into 'liposomes'".

Reviewer #3 (Remarks to the Author):

The manuscript by Su and colleagues describes novel x-ray structures of the RND efflux pump from *Campylobacter jejuni* named CmeB that has been captured in two distinct conformations. In one conformation each protomer of the trimer has a closed periplasmic funnel and in the other conformation one of these periplasmic funnels in the trimer are open. That the trimers can have asymmetric conformations is of no surprise, since it has been proposed for AcrB (a homologue of CmeB) there is cooperativity between the promoters. However, here they have reconstituted CmeB into liposomes and measured protein dynamics by FRET to shown that the different promoters seem to function independently and are therefore not cooperative. Very interesting.

This paper is of high-quality with lots of data and was a pleasure to read.

I have only a few minor comments.

1. The K843Cys mutant used for probe labelling is likely to be function since the double mutant K843-D409A does not show any clear FRET signals. However, its possible that the K843Cys mutant has poor activity compared to wildtype and it also possible that the double mutant is less stable. Could you please show at least gel filtration traces of the purified mutants compared to WT so that one at least knows that these mutants are well-folded? Ideally export data (i.e. in a complementation growth assay) of WT and the other mutants would be the best.

2. Please consider adding more labels (i.e. "open" or "closed") to your figures; its always nice when you can look a figure and understand the result of the experiment without having to read all the details. Also, I thought Fig. 6 was not a "simple model", but actually quite complicated to follow.

We are resubmitting a revised version of our manuscript entitled "Structures and transport dynamics of the *Campylobacter jejuni* multidrug efflux pump CmeB." Three of the referees have raised several important points that we have now addressed in this revision of the paper.

Our revised paper has an expanded scope, and several important sets of data have been added in response to the criticisms of the referees. We used the *C. jejuni* 81-176 $\Delta cmeABC::cat$ null mutant strain, which lacks the *cmeABC* genes. We inserted the *cmeABC* operon that includes *cmeABC*, *cmeR* and the intergenic region between *cmeR* and *cmeA* into the 16S region of the 81-176 genomic DNA.

One set of the additional data regards in vivo susceptibility assay of the mutant transporters. Western analysis suggested that the expression level of these mutant pumps are similar to that of wild-type CmeB. The results suggest that the K843C mutant is fully functional. We also examine the D409A mutant. We found that a single point mutation on this aspartate residue can abolish the function of the pump, suggesting that D409 is a critical residue.

In addition, we used isothermal titration calorimetry to define the binding affinity of bile acid to the CmeB pump. We found that the dissociation constant for taurodeoxycholate binding by CmeB is within the micromolar range.

Further, we made the CmeB protoliposomes and used an in vitro approach to investigate proton translocation across the membrane. We found that protons can be transferred into liposomes containing wild-type CmeB. When reconstituted the D409A mutant transporter into liposomes, this mutant pump did not translocate protons into the intravesicular space, confirming the importance of this residue for the function of the pump.

Additionally, we have also mutated K781 into a cysteine. This residue is located at the top of the funnel region (subdomain DC). According to our crystal structures, it was predicted that this region should not have any significant motions. Thus, we used these single-molecule FRET data as a negative control. FRET efficiency distributions histogram yielded a single narrow peak, indicating that this funnel region lacks major conformational changes throughout the efflux process.

Our point-by-point response to the referees' comments follows this letter.

This manuscript provides a direct evidence (for the first time) for the transport mechanism of RND-type multidrug efflux pumps. Thus, it has a high potential medical impact.

We believe that our paper is a state-of-the-art structure-function study of this important protein at the single-molecule level, and thus our work uncovers many important details of its mechanism that were not previously understood. We thank your reviewers that they have played in improving our manuscript to make the results more comprehensive and more comprehensible. They should be pleased with our responses to their important points and our manuscript ought now to be publishable.

Sincerely,

Referee #1

Structure:

- From Figure 1 and 2, it is very difficult to relate to the binding and extrusion states of the prototype AcrB, from where the designations come from. A defined table of the rmsd's of the PN1/PC2 subdomains, and PN2/PC1 subdomains, as well as the two parallel halves of the transmembrane domains must be given. Additionally, a figure with the differences between AcrB and CmeB differences in the regions with largest deviation should be given. This is also useful for the differences between the extrusion states in trimer form I and trimer form II (especially since a "new state" is mentioned).

We have made two new Tables (Tables S1 and S2) to compare different subdomains and two halves of the transmembrane domains. In addition, we have provided another new Table (Table S3) to compare the form I and form II structures of CmeB. New figures showing the differences between AcrB and CmeB (Figs. S3, S4 and S6) were also provided.

- The extrusion states in form I and form II appear different. Moreover, the calculated channels between the extrusion protomers in form I and form II are different. Are the channels at its narrowest points large enough for substrates to passage? In form I, the tilted helix from the PN1 subdomain appears to block the channel more than the extrusion state in form II, where the neighboring protomer is in the binding state and does not have a tilted helix (or does it?). Why does the green protomer ("new state") in fig. 2d not have an extrusion channel? What is different compared to the other extrusion state protomers?

The narrowest region of the extrusion channels in the form I and form II structures are 13.5 Å and 13.3 Å in diameter. These opening are wide enough for drug molecules to pass through. The corresponding diameter of the AcrB extrusion channel is 12.3 Å. A new figure (Fig. S4c) has been added to show the structure at the vicinity of the narrowest region of the resting state in this revised manuscript.

- Do all the extrusion protomers have the same transmembrane domain helix and side chain conformations (especially the Asp's and Lys at helix 4 and 10)?

All of the extrusion protomers have a similar conformation in the transmembrane domain (with rmsd ~0.4 Å). The conformation and position of side chains of residues D409, D410 and K935, which form a proton relay network, are also similar (Fig. S5).

- Which pdb of reference 14 has been used for MR? Why use a 2.8 Å AcrB model with modest geometry when a 1.9 Å structure (with better geometry) is available?

We have used the 1.9 Å-resolution structure of AcrB (PDB ID: 4DX5) for initial MR to determine our structure of CmeB. The citation of this AcrB paper has been added.

- The reported Rfree is very low considering the resolution given (especially for the P1 crystal). Very often the low Rfree still comes from the search model which has not been “shaken” in the refinement. Moreover, also the Rwork values are extremely low given the resolutions. Was the initial Rwork/Rfree after MR and the first cycle of refinement in the expected range (35-40%) and has this been consecutively lowered by model building and refinement rounds?

The initial Rwork/Rfree of the form I structure of CmeB after MR and first run of refinement was 34.9/42.0. For the form II structure, the initial refinement after MR resulted in a relatively high Rwork/Rfree 40.7/44.9. After several rounds of rebuilding and refinement, the Rfactors were lowered to the reported values. We analyzed 735 structures, in which the resolutions are around 3.6 Å, in the PDB databank. We found that the mean Rfree factor is 0.282. The Rfree factor reported in our P1 crystal is right at the center of the distribution (see figure below). Therefore, our Rfree is in a reasonable range according to statistic analysis.

- Could the entire CmeB molecule be assigned (in both crystal forms)?

Yes, we have assigned entire CmeB molecules in both forms, except the last seven residues at the C-terminus.

- A picture (close-up) of the distal binding pocket of CmeB should be provided (how different is it from AcrB). Could the binding protomer in principle bind drug and have co-crystallization trials been tried?

We have provided a new figure (Fig. S3) to indicate the distal binding pockets of CmeB and AcrB.

- What is the reason that form II is in the extrusion-extrusion-binding conformation? I guess that crystal contacts made with the binding protomer confer this asymmetry?

Yes, the crystal lattices are very different in these two forms. A new figure that shows the packing of these two crystal forms has been added in this revised version of the manuscript.

- Lines 136-140: Although I find the independent cycling of the protomers an interesting hypothesis, I don't think MdtBC is a good example for independent cycling. In fact, it is an example for dependent cycling, since two of the protomers are doing the energy transduction (and can't bind drugs), whereas the other is involved in drug binding.....(and can't transduce)(Kim et al., 2012). Unless the protomers are interacting in a concerted way, one cannot explain why the protomers incapable of energy transduction do expel the drugs (which is energy-dependent).

We have communicated with Professor Hiroshi Nikaido (the corresponding author of the MdtB₂C paper). He told us that our interpretation is correct. Within the heterotrimer, MdtB and MdtC function differently and that MdtC is likely to be responsible for recognizing and transporting drugs by itself.

Functional studies:

- In line 154 it is stated: "To elucidate if a CmeB protomer can export drugs individually within the trimer,....", however, despite FRET analysis will indicate dynamics of the transporter, it will not tell whether the reconstituted CmeB is actually transporting the drug (in this case Tdc).

We reconstituted the CmeB protein into proteoliposomes to perform single-molecule transport dynamics study. The data should reflect the actual drug transport process. Nonetheless, we have performed several additional experiments to demonstrate the function of CmeB. First, we used isothermal titration calorimetry (ITC) to show that the purified CmeB is capable of recognizing Tdc with the binding affinity at the low micromolar range, which is consistent with other RND protein. Secondly, we reconstituted the purified CmeB protein into proteoliposomes to perform in vitro proton translocation assay, indicating that protons can be transferred across the membrane of these CmeB proteoliposomes.

- The CmeB single Cys variants were made in a Cys-less background, where the cognate Cys residues have been replaced by Ser. Is this Cys-less variant (and the single substitution variant K834C) still active (MIC, fluorescent substrate extrusion e.g.)? This must be tested.

We have added an in vivo drug susceptibility experiment in this revised version of the manuscript. The data suggested that the cysteineless CmeB pump is fully functional.

- The pmf consist of two components (ΔpH and $\Delta \psi$). Both components should be tested alone and in combination in the assay and their effect on the dwelling times.

As Reviewer 1 mentioned, the pmf consists of two components (ΔpH and $\Delta\psi$). In this manuscript, we mainly focus on ΔpH , since RND family proteins are well characterized to be dependent on pH gradient. Technically, it is very difficult to test $\Delta\psi$ alone and in combination in the assay and their effect on the dwelling time. First, changing $\Delta\psi$ through Na^+ or K^+ gradient will alter ΔpH simultaneously as protons can be transferred across the membrane. Thus, these will make data very difficult to interpret. Secondly, these assays will take a huge amount of single-molecule experiment time and require a massive data analysis even if the experiment is working. The experiment is simply not feasible. Nonetheless, we have removed the term “PMF-dependent” and substituted it by “ ΔpH ” in this revision.

- I think it is important to test whether protons are transported across the membrane in the assay used. The use of pyranine dye might be helpful for this. Unless the proteoliposomes are leaky for protons (or maybe anions going inside the proteoliposomes, I would argue that the transported protons would create within seconds an opposite $\Delta\psi$ and stall further transport.

We totally agreed with the reviewer that it is important to test whether protons are able to transport across the membrane. We have performed key experiments suggested by this reviewer to test if protons can translocate across the membrane of the CmeB proteoliposomes using the pyranine dye. The results clearly show that CmeB can transport protons across the membrane.

- How stable is the ΔpH applied (should be tested)? Since measurements are done for up to 100s, is the ΔpH still present?

The results of assays show ΔpH is still present after 250s. Based on simple estimation, there are about one order of magnitude higher amount of trimeric CmeB molecules in these pyranine dye loaded proteoliposome than in single molecule experiment. Therefore, we assume it will take longer to completely deplete proton gradient in single molecule level.

- Without indication that protons are transported from outside to inside into the vesicles, data on conformational dynamics could also be interpreted as substrate binding, proton binding and their release to the same side where these solutes came from (the interproteoliposomal space), but no actual transport occurs (and hence no complete cycle of binding-extrusion-binding, but rather binding-release-binding). So the statement in lines 191-192 is almost correct: “Based on the traces and symmetrical nature of the density plots, it is more likely that the three protomers function independently of each other”. However, the question is: do they transport the drug (and protons)?

We have added two important sets of data (in vivo drug susceptibility and in vitro proton translocation) to show that CmeB mediates resistance to drugs and catalyzes transfer of protons across the membrane in this revised manuscript.

- How stable are the proteoliposomes in the presence of up to 10 μ M Taurodeoxycholate? Is there no interference with the integrity of the proteoliposomes? (Is the Δ pH affected?)

We have performed in vitro proteoliposome assays and demonstrated that the presence of 10 μ M taurodeoxycholate does not affect the integrity of the liposomes.

- I think another FRET pair should be tested as negative control, i.e. a pair which does not change its distance upon changing into the different conformational states (based on the structure).

The residue K781, which is located at the funnel domain of protein, was selected as a control. The results indeed confirm the robust performances of fluorophores and sm-FRET experiments.

- The frequency of transitions is reduced 7 times for the D409A variant. This appears not so much for a transporter which is incapable of transporting drugs and H⁺ (again, this has to be shown in a phenotypical assay for CmeB as well, otherwise it is just an assumption). I agree that this can be interpreted that certain states still can be probed by the transporter, but why have not only less states, but also slower transitions? Do protons still have binding capacity in the transporter? Would the transitions be completely vanished if a double variant (also replacing the second Asp) would be tested?

D409A variant shows not only 7 times reduced frequency of transitions but also the longer dwelling time of L \rightarrow I₁ and lack of the extrusion state. As the population level, we observed 95% of Low FRET at center of 0.2. We believe that such slow transition and without extrusion state, the D409A mutant is not able to export drug molecules. To prove this, we performed in vivo susceptibility assay to determine the function of D409A and D410A variants. The results indicate that both a single point mutation on D409 or D410 to an alanine can abolish the function of the pump. In addition, in vitro proton transport assay also shows the D409A mutant is not able to allow for the transport of protons cross the membrane. These two sets of new experimental data have been included in the revised version of this manuscript.

- I have a conceptual question on the description of the resting state. The resting state has its cleft close and theoretically no access path for drugs. If I understand it correctly, the opening and conversion into a binding state would be pmf dependent. Under physiological conditions, when does this occur? Cells always have pmf (unless they are dead), so why have a resting state? On the other hand, if drug would be a trigger (which seems at least intuitively logical), where is the drug binding site in the resting state with the closed cleft?

There are several intermediate states of the CmeB pump during the transport cycle. We have found the resting, binding and extrusion states. During drug export, the pump must go through all of these intermediate states to complete the transport cycle. We have observed the resting state of the pump both in the presence and absence of the PMF. However, it appears that this resting state

is much more stable in the absence of the PMF. The conformations of the resting and extrusion forms are significantly different and can be easily to separate into two different states. It also appears that the CmeB pump may not be ready to bind drug within the periplasmic drug binding site when it is in the resting state. To bind drug molecules, the CmeB pump may need to shift to the binding state. The extrusion state allows the pump to export drug and complete the transport cycle.

- Related to this, line 221 states that one of the transitions in the absence of drug is considered “resting-resting”. Nevertheless, in the proteoliposomal FRET experiment, a ΔpH is present. If the resting state is only in the absence of pmf, how is this possible (or I misunderstood the definition of “pmf-dependent”).

The majority transitions in the absence of drug are between resting and binding state, which correspond to the FRET observations of “resting-resting” to “resting-binding” or “resting-binding” to “resting-resting”. In fact, the resting state can be found both in the presence and absence of PMF. Since the transitions are reversible, we also observe the resting state even in the presence of PMF. However, it appears that the resting state is much more stable when there is no PMF.

- In lines 248-250, the statement couples the presence of pmf and substrate to be triggers for resting to binding conversion. If both are necessary, the question remains when the cell lacks the pmf in any stage of its life?

This is not the focus of our research. We would like to point out that there are several intermediate states through the transport cycle of the CmeB pump. In this case, we have identified the resting, binding and extrusion states of the pump. The alteration between different states may need to couple with the transfer of protons. The resting state is one of the intermediates that the pump must go through during drug export. It appears that the resting state of the pump is much more stable in the absence of proton transfer.

- A technical question: Why is cyclooctatetraene (2 mM) added to the imaging experiments. Could it be a substrate of the pump as well and has this been checked?

Cyclooctatetraene (Cot) is an efficient triplet state quencher that reduces the lifetime of dark states. To address the concern that Cot could be a substrate of CmeB, we measured the binding affinity of Tdc to CmeB both in the presence and absence of 2mM Cot. The results indicate that the presence of Cot has no significant effect on the binding of Tdc to the CmeB pump.

- Lines 400-406, what was the % of the total proteoliposomes (traces) which did match with the four criteria?

Approximately 10% of molecules match these criteria.

Referee #2

The manuscript by Su et al. describes a model for the transport mechanism of the subunits of the CmeB trimer by proposing that the subunits function independently rather than in a cooperative manner. Given that CmeB is an RND-type multi-substrate efflux transporter, which contributes to bacterial resistance to a range of structurally unrelated antibiotic compounds, understanding structural dynamics of this transporter is important since it may contribute towards development of novel compounds against multidrug resistant *C. jejuni*. The manuscript documents the use of a number of complementary experimental techniques including X-ray crystallography, single-molecule FRET (smFRET) spectroscopy and Hidden Markov modeling (HMM) to support the proposed model. While the model is interesting I have several comments that are detailed below:

1. On p. 7, lines 128-135 the authors write: "Our structures of CmeB ... that individual protomers of these trimeric RND pumps could bind and export substrates independently instead of operating in a synchronized fashion." If the individual protomers can function independently why do they have to assemble into trimers? Why not monomers or pentamers, for example? Please comment.

Thus far, all known HAE-RND pumps are found to be trimeric in form (homo- or hetero-trimer). This suggests that the trimer may be the most stable oligomerization state for these membrane proteins.

2. p. 8, line 166: Given that the sm-FRET data do not agree with the proposed rotating mechanism what is the evidence that replacing K843 to a cysteine for the attachment of fluorophores does not alter the function of CmeB?

We have added an in vivo drug susceptibility experiment in this revised version of the manuscript. Our data indicate that the K843C mutant is fully functional.

3. Transition density plots in Fig 4c and Fig. 5a should be fitted with Gaussians. The number of photobleaching steps, which tells the number of fluorophores attached to a trimer, does not seem to correspond to the number of Gaussians that would be expected in Fig 4c (e.g. top) and 5a (e.g. bottom). The bin size was set at 0.02 (p. 19, line 412). Given that the peaks of the distributions can be affected with different bin sizes and the selection of the bin size is subjective, the proper way for selection of the bin size is by using maximum-likelihood estimation for peak fitting.

We thank the reviewer for this helpful suggestion. Fig. 4c, Fig. 5a and Fig. S15c-d histograms have now been fitted with Gaussians. Only those CmeB proteoliposomes that contain a single donor and an acceptor dyes FRET pairs were selected to compile FRET histograms in Fig4c,

Fig5a and Fig S15c-e. Therefore, the number of Gaussians in the histogram corresponds to the number of “states”. To clarify this, we have added more details in “Single-molecule FRET data analysis” section. We agree with the reviewer that the selection of bin size is subjective. To avoid bias, maximum-likelihood estimate (MLE) was used to fit the K781C data (Fig. S15c-e). After the peak fitting, we adjusted the bin size and found histograms with the bin size value of 0.02, which matches with the fitted peaks very well. In Fig. 4c, we used the peak values suggested from hidden Markov model (HHM) to fit the histograms. HHM has an objective state-finding algorithm, which is especially helpful for complicated transitions between multiple states. It provides us with unbiased estimated peak values (McKinney, Joo & Ha, *Biophys. J.*, 91:1941-1951 (2006)).

4. FRET between neighboring CmeB trimers on the same liposome could be a problem because of inter-molecular FRET. What percentage of labeled vs. unlabeled trimers was used for reconstitution in liposomes? Justify.

We have added a detailed discussion in the section of “Protein reconstitution into liposomes” to justify it in this revised version of the manuscript.

5. Which precaution was taken to avoid the effect of the finite size and orientation of fluorescent probes bringing additional difficulty to converting FRET measurements to the estimation of distances?

We have noticed that the difficulty of converting FRET measurement to the estimation of distance due to the finite size and length of the fluorophore. Therefore, our strategy is to focus on the relative distance changes of protein domains instead of measuring the absolute distances. As a control, we chose the residue K781 in each CmeB protomer. Based on our crystal structures, the vicinity of this residue is not supposed to have significant movements. We have included these data on p.10 of this revised manuscript.

6. p. 16, line 336: “Protein reconstitution into ‘liposomes’”.

The typo “liposomes” has been corrected.

Referee #3

The manuscript by Su and colleagues describes novel x-ray structures of the RND efflux pump from *Campylobacter jejuni* named CmeB that has been captured in two distinct conformations. In one conformation each protomer of the trimer has a closed periplasmic funnel and in the other conformation one of these periplasmic funnels in the trimer are open. That the trimers can have asymmetric conformations is of no surprise, since it has been proposed for AcrB (a homologue of CmeB) there is cooperativity between the promoters. However, here they have reconstituted CmeB into liposomes and measured protein dynamics by FRET to shown that the different

promoters seem to function independently and are therefore not cooperative. Very interesting.

This paper is of high-quality with lots of data and was a pleasure to read.

I have only a few minor comments.

1. The K843Cys mutant used for probe labelling is likely to be function since the double mutant K843-D409A does not show any clear FRET signals. However, its possible that the K843Cys mutant has poor activity compared to wildtype and it also possible that the double mutant is less stable. Could you please show at least gel filtration traces of the purified mutants compared to WT so that one at least knows that these mutants are well-folded? Ideally export data (i.e. in a complementation growth assay) of WT and the other mutants would be the best.

We have used in vivo drug susceptibility assay to show that the K843C mutant is fully functional in this revised version of the manuscript.

2. Please consider adding more labels (i.e. “open” or “closed”) to your figures; its always nice when you can look a figure and understand the result of the experiment without having to read all the details. Also, I thought Fig. 6 was not a “simple model”, but actually quite complicated to follow.

More labels have been added to the figures to make them easier to understand, following this reviewer’s suggestion. The word “simple” has been removed from the text in this revised version of the manuscript.

Reviewers' Comments:

Reviewer #1:

Remarks to the Author:

Referee #1

Structure:

1. From Figure 1 and 2, it is very difficult to relate to the binding and extrusion states of the prototype AcrB, from where the designations come from. A defined table of the rmsd's of the PN1/PC2 subdomains, and PN2/PC1 subdomains, as well as the two parallel halves of the transmembrane domains must be given. Additionally, a figure with the differences between AcrB and CmeB differences in the regions with largest deviation should be given. This is also useful for the differences between the extrusion states in trimer form I and trimer form II (especially since a "new state" is mentioned).

We have made two new Tables (Tables S1 and S2) to compare different subdomains and two halves of the transmembrane domains. In addition, we have provided another new Table (Table S3) to compare the form I and form II structures of CmeB. New figures showing the differences between AcrB and CmeB (Figs. S3, S4 and S6) were also provided.

Referee #1: These tables have improved the insights a lot.

2. The extrusion states in form I and form II appear different. Moreover, the calculated channels between the extrusion protomers in form I and form II are different. Are the channels at its narrowest points large enough for substrates to passage? In form I, the tilted helix from the PN1 subdomain appears to block the channel more than the extrusion state in form II, where the neighboring protomer is in the binding state and does not have a tilted helix (or does it?). Why does the green protomer ("new state") in fig. 2d not have an extrusion channel? What is different compared to the other extrusion state protomers?

The narrowest region of the extrusion channels in the form I and form II structures are 13.5 Å and 13.3 Å in diameter. These opening are wide enough for drug molecules to pass through. The corresponding diameter of the AcrB extrusion channel is 12.3 Å. A new figure (Fig. S4c) has been added to show the structure at the vicinity of the narrowest region of the resting state in this revised manuscript.

Referee #1: Great, OK

3. Do all the extrusion protomers have the same transmembrane domain helix and side chain conformations (especially the Asp's and Lys at helix 4 and 10)?

All of the extrusion protomers have a similar conformation in the transmembrane domain (with rmsd ~0.4 Å). The conformation and position of side chains of residues D409, D410 and K935, which form a proton relay network, are also similar (Fig. S5).

Referee #1: OK.

4. Which pdb of reference 14 has been used for MR? Why use a 2.8 Å AcrB model with modest geometry when a 1.9 Å structure (with better geometry) is available?

We have used the 1.9 Å-resolution structure of AcrB (PDB ID: 4DX5) for initial MR to determine our structure of CmeB. The citation of this AcrB paper has been added.

Referee #1: OK.

5. The reported Rfree is very low considering the resolution given (especially for the P1 crystal).

Very often the low Rfree still comes from the search model which has not been “shaken” in the refinement. Moreover, also the Rwork values are extremely low given the resolutions. Was the initial Rwork/Rfree after MR and the first cycle of refinement in the expected range (35-40%) and has this been consecutively lowered by model building and refinement rounds?

The initial Rwork/Rfree of the form I structure of CmeB after MR and first run of refinement was 34.9/42.0. For the form II structure, the initial refinement after MR resulted in a relatively high Rwork/Rfree 40.7/44.9. After several rounds of rebuilding and refinement, the Rfactors were lowered to the reported values. We analyzed 735 structures, in which the resolutions are around 3.6 Å, in the PDB databank. We found that the mean Rfree factor is 0.282. The Rfree factor reported in our P1 crystal is right at the center of the distribution (see figure below). Therefore, our Rfree is in a reasonable range according to statistic analysis.

Referee #1: What can I say? I can't beat statistics, but my crystallography teachers told me that you can't have a low Rfree model with low resolution data. Structures with 3.5 Å might have Rfree's of around 0.35, maybe up to 0.3 (so I was told). If you obtain higher resolution data (say, 2.5 Å), how far down does the Rfree get if it is already at 0.25? And certainly, the model does get much better from refining against 2.5 Å in comparison with 3.5 Å data. However, if the statistics above tell differently, I must be wrong.

6. Could the entire CmeB molecule be assigned (in both crystal forms)?

Yes, we have assigned entire CmeB molecules in both forms, except the last seven residues at the C-terminus.

Referee #1: OK.

7. A picture (close-up) of the distal binding pocket of CmeB should be provided (how different is it from AcrB). Could the binding protomer in principle bind drug and have co-crystallization trials been tried?

We have provided a new figure (Fig. S3) to indicate the distal binding pockets of CmeB and AcrB.

Referee #1: Great, OK.

8. What is the reason that form II is in the extrusion-extrusion-binding conformation? I guess that crystal contacts made with the binding protomer confer this asymmetry?

Yes, the crystal lattices are very different in these two forms. A new figure that shows the packing of these two crystal forms has been added in this revised version of the manuscript.

Referee #1: OK!

9. Lines 136-140: Although I find the independent cycling of the protomers an interesting hypothesis, I don't think MdtBC is a good example for independent cycling. In fact, it is an example for dependent cycling, since two of the protomers are doing the energy transduction (and can't bind drugs), whereas the other is involved in drug binding.....(and can't transduce)(Kim et al., 2012). Unless the protomers are interacting in a concerted way, one cannot explain why the protomers incapable of energy transduction do expel the drugs (which is energy-dependent).

We have communicated with Professor Hiroshi Nikaido (the corresponding author of the MdtB2C paper). He told us that our interpretation is correct. Within the heterotrimer, MdtB and MdtC function differently and that MdtC is likely to be responsible for recognizing and transporting drugs by itself.

Referee #1: If independent cycling means that every protomer binds/transport drugs and the energy conversion is also within the same protomer (so every protomer just goes through a

cycling Resting/Binding/Extrusion no matter in what state the other protomers are), how can this be valid for MdtBC? MdtB cannot bind drugs, how can it independently transport drugs? MdtC can only bind drugs but cannot energize the transport process, how can it transport the bound drug? The latter can only transport the drug if the energy conversion from proton motive force into conformational change in MdtB is transduced to MdtC. So MdtC is dependent on MdtB. Unless the authors mean with "independent cycling" something different, I really don't understand the comparison.

Functional studies:

10. In line 154 it is stated: "To elucidate if a CmeB protomer can export drugs individually within the trimer,....", however, despite FRET analysis will indicate dynamics of the transporter, it will not tell whether the reconstituted CmeB is actually transporting the drug (in this case Tde).

We reconstituted the CmeB protein into proteoliposomes to perform single-molecule transport dynamics study. The data should reflect the actual drug transport process. Nonetheless, we have performed several additional experiments to demonstrate the function of CmeB. First, we used isothermal titration calorimetry (ITC) to show that the purified CmeB is capable of recognizing Tdc with the binding affinity at the low micromolar range, which is consistent with other RND protein. Secondly, we reconstituted the purified CmeB protein into proteoliposomes to perform in vitro proton translocation assay, indicating that protons can be transferred across the membrane of these CmeB proteoliposomes.

Referee #1: I think it is great the authors did the experiments (which show important additional detail) and I will not make any other issue out of it.

I would like to state though that: a. binding does not imply transport and b. proton movement is a strong suggestion that transport might happen, but does not necessarily imply the coupling to transport (for that a simultaneous Tde transport has to be shown).

11. The CmeB single Cys variants were made in a Cys-less background, where the cognate Cys residues have been replaced by Ser. Is this Cys-less variant (and the single substitution variant K834C) still active (MIC, fluorescent substrate extrusion e.g.)? This must be tested.

We have added an in vivo drug susceptibility experiment in this revised version of the manuscript. The data suggested that the cysteineless CmeB pump is fully functional.

Referee #1: Very important data. Great.

12. The pmf consist of two components (ΔpH and $\Delta \psi$). Both components should be tested alone and in combination in the assay and their effect on the dwelling times.

As Reviewer 1 mentioned, the pmf consists of two components (ΔpH and $\Delta \psi$). In this manuscript, we mainly focus on ΔpH , since RND family protein are well characterized to be dependence of pH gradient. Technically, it is very difficult to test $\Delta \psi$ alone and in combination in the assay and their effect on the dwelling time. First, changing $\Delta \psi$ through Na^+ or K^+ gradient will alter ΔpH simultaneously as protons can be transferred across the membrane. Thus, these will make data very difficult to interpret. Secondly, these assays will take a huge amount of single-molecule experiment time and require a massive data analysis even if the experiment is working. The experiment is simply not feasible. Nonetheless, we have removed the term "PMF-dependent" and substituted it by " ΔpH " in this revision.

Referee #1: A fair compromise, and I understand the difficulties of passive H^+ influx with proteoliposomes upon application of a negative $\Delta \psi$. However, many transport assays have been conducted in proteoliposomes (with other transporters) and $\Delta \psi$ -dependent transport is not impossible but are indeed reliant on proton-tight proteoliposomes.

13. I think it is important to test whether protons are transported across the membrane in the assay used. The use of pyranine dye might be helpful for this. Unless the proteoliposomes are leaky for protons (or maybe anions going inside the proteoliposomes), I would argue that the transported protons would create within seconds an opposite delta psi and stall further transport.

We totally agreed with the reviewer that it is important to test whether protons are able to transport across the membrane. We have performed key experiments suggested by this reviewer to test if protons can translocate cross the membrane of the CmeB proteoliposomes using the pyranine dye. The results clearly show that CmeB can transport protons across the membrane.

Referee #1: Nice experiment. Was this experiment also done with valinomycin present (and equimolar K⁺ on both sides of the membrane)? This should in principle enhance the signal (quenching) because more protons can enter the proteoliposomes (since proton entry will lead to an "inverse" delta_psi and stall further influx).

14. How stable is the delta_pH applied (should be tested)? Since measurements are done for up to 100s, is the delta_pH still present?

The results of assays show delta_pH is still present after 250s. Based on simple estimation, there are about one order of magnitude higher amount of trimeric CmeB molecules in these pyranine dye loaded proteoliposome than in single molecule experiment. Therefore, we assume it will take longer to completely deplete proton gradient in single molecule level.

Referee #1: OK.

15. Without indication that protons are transported from outside to inside into the vesicles, data on conformational dynamics could also be interpreted as substrate binding, proton binding and their release to the same side where these solutes came from (the interproteoliposomal space), but no actual transport occurs (and hence no complete cycle of binding-extrusion-binding, but rather binding-release-binding). So the statement in lines 191-192 is almost correct: "Based on the traces and symmetrical nature of the density plots, it is more likely that the three protomers function independently of each other". However, the question is: do they transport the drug (and protons)?

We have added two important sets of data (in vivo drug susceptibility and in vitro proton translocation) to show that CmeB mediates resistance to drugs and catalyzes transfer of protons across the membrane in this revised manuscript.

Referee #1: Yes, but also see above. In vivo resistance is mediated by a tripartite system, not by CmeB itself. The observation in whole cells cannot be translated to your proteoliposome setup with only CmeB present. Transport of protons is, as mentioned above, not necessarily telling that taurocholate is transported by CmeB (alone). I refer here to the work on in vitro tripartite setup (two batch liposomes) of MexAB-OprM by the laboratory of Martin Picard.

I am not suggesting to do that particular experiment, but would like to suggest to be careful with a sentence in line 68/69 (also similar wording in line 162):

"Using sm-FRET imaging, we demonstrate that each CmeB protomer within the trimer is able to export drugs independently", since you do not demonstrate drug transport. You demonstrate possible drug binding via ITC and possible H⁺ influx in the presence of CmeB, which is not present if you use CmeB_D409A. From this data, the interpretation could be "CmeB transport drugs", if you assume it to be coupled to the observed H⁺ transport.

16. How stable are the proteoliposomes in the presence of up to 10 uM Taurodeoxycholate? Is there no interference with the integrity of the proteoliposomes? (Is the delta_pH affected?)

We have performed in vitro proteoliposome assays and demonstrated that the presence of 10 μM taurodeoxycholate does not affect the integrity of the liposomes.

Referee #1: OK.

17. I think another FRET pair should be tested as negative control, i.e. a pair which does not change its distance upon changing into the different conformational states (based on the structure).

The residue K781, which is located at the funnel domain of protein, was selected as a control. The results indeed confirm the robust performances of fluorophores and sm-FRET experiments.

Referee #1: Very important control. OK.

18. The frequency of transitions is reduced 7 times for the D409A variant. This appears not so much for a transporter which is incapable of transporting drugs and H^+ (again, this has to be shown in a phenotypical assay for CmeB as well, otherwise it is just an assumption). I agree that this can be interpreted that certain states still can be probed by the transporter, but why have not only less states, but also slower transitions? Do protons still have binding capacity in the transporter? Would the transitions be completely vanished if a double variant (also replacing the second Asp) would be tested?

D409A variant shows not only 7 times reduced frequency of transitions but also the longer dwelling time of $L \leftrightarrow I1$ and lack of the extrusion state. As the population level, we observed 95% of Low FRET at center of 0.2. We believe that such slow transition and without extrusion state, the D409A mutant is not able to export drug molecules. To prove this, we performed in vivo susceptibility assay to determine the function of D409A and D410A variants. The results indicate that both a single point mutation on D409 or D410 to an alanine can abolish the function of the pump. In addition, in vitro proton transport assay also shows the D409A mutant is not able to allow for the transport of protons cross the membrane. These two sets of new experimental data have been included in the revised version of this manuscript.

Referee #1: OK.

19. I have a conceptual question on the description of the resting state. The resting state has its cleft closed and theoretically no access path for drugs. If I understand it correctly, the opening and conversion into a binding state would be pmf dependent. Under physiological conditions, when does this occur? Cells always have pmf (unless they are dead), so why have a resting state? On the other hand, if drug would be a trigger (which seems at least intuitively logical), where is the drug binding site in the resting state with the closed cleft?

There are several intermediate states of the CmeB pump during the transport cycle. We have found the resting, binding and extrusion states. During drug export, the pump must go through all of these intermediate states to complete the transport cycle. We have observed the resting state of the pump both in the presence and absence of the PMF. However, it appears that this resting state is much more stable in the absence of the PMF. The conformations of the resting and extrusion forms are significantly different and can be easily to separate into two different states. It also appears that the CmeB pump may not be ready to bind drug within the periplasmic drug binding site when it is in the resting state. To bind drug molecules, the CmeB pump may need to shift to the binding state. The extrusion state allows the pump to export drug and complete the transport cycle.

Referee #1: Above is all true. But why have a resting state? Especially if the protomers cycle independently, only binding and extrusion states seem necessary. There appears no physiological need for a resting state (in the presence of a pmf). I still am puzzled by this observation, and in

the light of independent cycling there seems to be no obvious interpretation.

20. Related to this, line 221 states that one of the transitions in the absence of drug is considered "resting-resting". Nevertheless, in the proteoliposomal FRET experiment, a Δ pH is present. If the resting state is only present in the absence of pmf, how is this possible (or I misunderstood the definition of "pmf-dependent").

The majority transitions in the absence of drug are between resting and binding state, which correspond to the FRET observations of "resting-resting" to "resting-binding" or "resting-binding" to "resting-resting". In fact, the resting state can be found both in the presence and absence of PMF. Since the transitions are reversible, we also observe the resting state even in the presence of PMF. However, it appears that the resting state is much more stable when there is no PMF.

Referee #1: OK (but see below).

21. In lines 248-250, the statement couples the presence of pmf and substrate to be triggers for resting to binding conversion. If both are necessary, the question remains when the cell lacks the pmf in any stage of its life?

This is not the focus of our research. We would like to point out that there are several intermediate states through the transport cycle of the CmeB pump. In this case, we have identified the resting, binding and extrusion states of the pump. The alteration between different states may need to couple with the transfer of protons. The resting state is one of the intermediates that the pump must go through during drug export. It appears that the resting state of the pump is much more stable in the absence of proton transfer.

Referee #1: This might not be the focus of your research, however, you made this statement based on your observations and of course it has to translate to the physiological role of the pump states, otherwise why do the research? If the authors state that the pmf is a trigger, besides substrates, I think the question of when the pmf can be a trigger in a living cell (where the pmf is constantly present (or is it?)) is warranted. Btw. your answer above is not even trying to interpret that result, it only describes the results you obtained.

22. A technical question: Why is cyclooctatetraene (2 mM) added to the imaging experiments. Could it be a substrate of the pump as well and has this been checked?

Cyclooctatetraene (Cot) is an efficient triplet state quencher that reduces the lifetime of dark states. To address the concern that Cot could be a substrate of CmeB, we measured the binding affinity of Tdc to CmeB both in the presence and absence of 2mM Cot. The results indicate that the presence of Cot has no significant effect on the binding of Tdc to the CmeB pump.

Referee #1: OK, but it shows (although I have not seen the results of the experiment, I have to believe the statement above) only that Cot is not directly competing with Tdc. RND pumps appear to have multiple binding sites (Yu et al., 2003, Yu et al., 2005, Murakami et al., 2006, Nakashima et al., 2011, Eicher et al., 2012, Oswald et al., 2016).

23. Lines 400-406, what was the % of the total proteoliposomes (traces) which did match with the four criteria?

Approximately 10% of molecules match these criteria.

Referee #1: OK.

Reviewer #2:

Remarks to the Author:

In their revised manuscript The authors have adequately and in a satisfactory manner addressed my critique of the original article. I recommend the revised manuscript for publication in Nature Communications in its present form.

Reviewer #3:

Remarks to the Author:

I am more than satisfied with the responses to my previous enquires. A great paper!

POINT-BY-POINT RESPONSE TO REFREES:

REVIEWERS' COMMENTS:

Reviewer #1 (Remarks to the Author):

Referee #1

Structure:

1. From Figure 1 and 2, it is very difficult to relate to the binding and extrusion states of the prototype AcrB, from where the designations come from. A defined table of the rmsd's of the PN1/PC2 subdomains, and PN2/PC1 subdomains, as well as the two parallel halves of the transmembrane domains must be given. Additionally, a figure with the differences between AcrB and CmeB differences in the regions with largest deviation should be given. This is also useful for the differences between the extrusion states in trimer form I and trimer form II (especially since a "new state" is mentioned).

We have made two new Tables (Tables S1 and S2) to compare different subdomains and two halves of the transmembrane domains. In addition, we have provided another new Table (Table S3) to compare the form I and form II structures of CmeB. New figures showing the differences between AcrB and CmeB (Figs. S3, S4 and S6) were also provided.

Referee #1: These tables have improved the insights a lot.

OK

2. The extrusion states in form I and form II appear different. Moreover, the calculated channels between the extrusion protomers in form I and form II are different. Are the channels at its narrowest points large enough for substrates to passage? In form I, the tilted helix from the PN1 subdomain appears to block the channel more than the extrusion state in form II, where the neighboring protomer is in the binding state and does not have a tilted helix (or does it?). Why does the green protomer ("new state") in fig. 2d not have an extrusion channel? What is different compared to the other extrusion state protomers?

The narrowest region of the extrusion channels in the form I and form II structures are 13.5 Å and 13.3 Å in diameter. These opening are wide enough for drug molecules to pass through. The corresponding diameter of the AcrB extrusion channel is 12.3 Å. A new figure (Fig. S4c) has been added to show the structure at the vicinity of the narrowest region of the resting state in this revised manuscript.

Referee #1: Great, OK

OK

3. Do all the extrusion protomers have the same transmembrane domain helix and side chain conformations (especially the Asp's and Lys at helix 4 and 10)?

All of the extrusion protomers have a similar conformation in the transmembrane domain (with rmsd ~0.4 Å). The conformation and position of side chains of residues D409, D410 and K935, which form a proton relay network, are also similar (Fig. S5).

Referee #1: OK.

OK

4. Which pdb of reference 14 has been used for MR? Why use a 2.8 Å AcrB model with modest geometry when a 1.9 Å structure (with better geometry) is available?

We have used the 1.9 Å-resolution structure of AcrB (PDB ID: 4DX5) for initial MR to determine our structure of CmeB. The citation of this AcrB paper has been added.

Referee #1: OK.

OK

5. The reported Rfree is very low considering the resolution given (especially for the P1 crystal). Very often the low Rfree still comes from the search model which has not been “shaken” in the refinement. Moreover, also the Rwork values are extremely low given the resolutions. Was the initial Rwork/Rfree after MR and the first cycle of refinement in the expected range (35-40%) and has this been consecutively lowered by model building and refinement rounds?

The initial Rwork/Rfree of the form I structure of CmeB after MR and first run of refinement was 34.9/42.0. For the form II structure, the initial refinement after MR resulted in a relatively high Rwork/Rfree 40.7/44.9. After several rounds of rebuilding and refinement, the Rfactors were lowered to the reported values. We analyzed 735 structures, in which the resolutions are around 3.6 Å, in the PDB databank. We found that the mean Rfree factor is 0.282. The Rfree factor reported in our P1 crystal is right at the center of the distribution (see figure below). Therefore, our Rfree is in a reasonable range according to statistic analysis.

Referee #1: What can I say? I can't beat statistics, but my crystallography teachers told me that you can't have a low Rfree model with low resolution data. Structures with 3.5 A might have Rfree's of around 0.35, maybe up to 0.3 (so I was told). If you obtain higher resolution data (say, 2.5 A), how far down does the Rfree get if it is already at 0.25? And certainly, the model does get much better from refining against 2.5 A in comparison with 3.5 A data. However, if the statistics above tell differently, I must be wrong.

OK

6. Could the entire CmeB molecule be assigned (in both crystal forms)?

Yes, we have assigned entire CmeB molecules in both forms, except the last seven residues at the C-terminus.

Referee #1: OK.

OK

7. A picture (close-up) of the distal binding pocket of CmeB should be provided (how different is it from AcrB). Could the binding protomer in principle bind drug and have co-crystallization trials been tried?

We have provided a new figure (Fig. S3) to indicate the distal binding pockets of CmeB and AcrB.

Referee #1: Great, OK.

OK

8. What is the reason that form II is in the extrusion-extrusion-binding conformation? I guess that crystal contacts made with the binding protomer confer this asymmetry?

Yes, the crystal lattices are very different in these two forms. A new figure that shows the packing of these two crystal forms has been added in this revised version of the manuscript.

Referee #1: OK!

OK

9. Lines 136-140: Although I find the independent cycling of the protomers an interesting hypothesis, I don't think MdtBC is a good example for independent cycling. In fact, it is an example for dependent cycling, since two of the protomers are doing the energy transduction (and can't bind drugs), whereas the other is involved in drug binding.....(and can't transduce)(Kim et al., 2012). Unless the protomers are interacting in a concerted way, one

cannot explain why the protomers incapable of energy transduction do expel the drugs (which is energy-dependent). We have communicated with Professor Hiroshi Nikaido (the corresponding author of the MdtB₂C paper). He told us that our interpretation is correct. Within the heterotrimer, MdtB and MdtC function differently and that MdtC is likely to be responsible for recognizing and transporting drugs by itself.

Referee #1: If independent cycling means that every protomer binds/transport drugs and the energy conversion is also within the same protomer (so every protomer just goes through a cycling Resting/Binding/Extrusion no matter in what state the other protomers are), how can this be valid for MdtBC? MdtB cannot bind drugs, how can it independently transport drugs? MdtC can only bind drugs but cannot energize the transport process, how can it transport the bound drug? The latter can only transport the drug if the energy conversion from proton motive force into conformational change in MdtB is transduced to MdtC. So MdtC is dependent on MdtB. Unless the authors mean with “independent cycling” something different, I really don't understand the comparison.

MdtC itself has its own proton relay network. The conserved residues D401, D402 and K919 of MdtC make up its own proton relay triad in the transmembrane region. We think that MdtC can bind and export its own substrates using its own energy source. Nonetheless, the statements “This is evidenced through the heterotrimeric MdtB₂C efflux pump, where the MdtB and MdtC subunits function differently. It was found that MdtC is likely to be responsible for recognizing and transporting drugs by itself within the heterotrimeric efflux pump” have been removed from the manuscript. In addition, the citation of the MdtB₂C paper has been deleted in this revised version of the manuscript.

Functional studies:

10. In line 154 it is stated: “To elucidate if a CmeB protomer can export drugs individually within the trimer,...”, however, despite FRET analysis will indicate dynamics of the transporter, it will not tell whether the reconstituted CmeB is actually transporting the drug (in this case Tdc).

We reconstituted the CmeB protein into proteoliposomes to perform single-molecule transport dynamics study. The data should reflect the actual drug transport process. Nonetheless, we have performed several additional experiments to demonstrate the function of CmeB. First, we used isothermal titration calorimetry (ITC) to show that the purified CmeB is capable of recognizing Tdc with the binding affinity at the low micromolar range, which is consistent with other RND protein. Secondly, we reconstituted the purified CmeB protein into proteoliposomes to perform in vitro proton translocation assay, indicating that protons can be transferred across the membrane of these CmeB proteoliposomes.

Referee #1: I think it is great the authors did the experiments (which show important additional detail) and I will not make any other issue out of it.

I would like to state though that: a. binding does not imply transport and b. proton movement is a strong suggestion that transport might happen, but does not necessarily imply the coupling to transport (for that a simultaneous Tdc transport has to be shown).

OK

11. The CmeB single Cys variants were made in a Cys-less background, where the cognate Cys residues have been replaced by Ser. Is this Cys-less variant (and the single substitution variant K834C) still active (MIC, fluorescent substrate extrusion e.g.)? This must be tested.

We have added an in vivo drug susceptibility experiment in this revised version of the manuscript. The data suggested that the cysteineless CmeB pump is fully functional.

Referee #1: Very important data. Great.

OK

12. The pmf consist of two components (ΔpH and $\Delta \psi$). Both components should be tested alone and in combination in the assay and their effect on the dwelling times.

As Reviewer 1 mentioned, the pmf consists of two components (ΔpH and $\Delta\psi$). In this manuscript, we mainly focus on ΔpH , since RND family protein are well characterized to be dependence of pH gradient. Technically, it is very difficult to test $\Delta\psi$ alone and in combination in the assay and their effect on the dwelling time. First, changing $\Delta\psi$ through Na^+ or K^+ gradient will alter ΔpH simultaneously as protons can be transferred across the membrane. Thus, these will make data very difficult to interpret. Secondly, these assays will take a huge amount of single-molecule experiment time and require a massive data analysis even if the experiment is working. The experiment is simply not feasible. Nonetheless, we have removed the term “PMF-dependent” and substituted it by “ ΔpH ” in this revision.

Referee #1: A fair compromise, and I understand the difficulties of passive H^+ influx with proteoliposomes upon application of a negative $\Delta\psi$. However, many transport assays have been conducted in proteoliposomes (with other transporters) and $\Delta\psi$ -dependent transport is not impossible but are indeed reliant on proton-tight proteoliposomes.

OK

13. I think it is important to test whether protons are transported across the membrane in the assay used. The use of pyranine dye might be helpful for this. Unless the proteoliposomes are leaky for protons (or maybe anions going inside the proteoliposomes), I would argue that the transported protons would create within seconds an opposite $\Delta\psi$ and stall further transport.

We totally agreed with the reviewer that it is important to test whether protons are able to transport across the membrane. We have performed key experiments suggested by this reviewer to test if protons can translocate cross the membrane of the CmeB proteoliposomes using the pyranine dye. The results clearly show that CmeB can transport protons across the membrane.

Referee #1: Nice experiment. Was this experiment also done with valinomycin present (and equimolar K^+ on both sides of the membrane)? This should in principle enhance the signal (quenching) because more protons can enter the proteoliposomes (since proton entry will lead to an “inverse” $\Delta\psi$ and stall further influx).

OK

14. How stable is the ΔpH applied (should be tested)? Since measurements are done for up to 100s, is the ΔpH still present?

The results of assays show ΔpH is still present after 250s. Based on simple estimation, there are about one order of magnitude higher amount of trimeric CmeB molecules in these pyranine dye loaded proteoliposome than in single molecule experiment. Therefore, we assume it will take longer to completely deplete proton gradient in single molecule level.

Referee #1: OK.

OK

15. Without indication that protons are transported from outside to inside into the vesicles, data on conformational dynamics could also be interpreted as substrate binding, proton binding and their release to the same side where these solutes came from (the interproteoliposomal space), but no actual transport occurs (and hence no complete cycle of binding-extrusion-binding, but rather binding-release-binding). So the statement in lines 191-192 is almost correct: “Based on the traces and symmetrical nature of the density plots, it is more likely that the three protomers function independently of each other”. However, the question is: do they transport the drug (and protons)?

We have added two important sets of data (in vivo drug susceptibility and in vitro proton translocation) to show that CmeB mediates resistance to drugs and catalyzes transfer of protons across the membrane in this revised manuscript.

Referee #1: Yes, but also see above. In vivo resistance is mediated by a tripartite system, not by CmeB itself. The

observation in whole cells cannot be translated to your proteoliposome setup with only CmeB present. Transport of protons is, as mentioned above, not necessarily telling that taurocholate is transported by CmeB (alone). I refer here to the work on in vitro tripartite setup (two batch liposomes) of MexAB-OprM by the laboratory of Martin Picard. I am not suggesting to do that particular experiment, but would like to suggest to be careful with a sentence in line 68/69 (also similar wording in line 162):

“Using sm-FRET imaging, we demonstrate that each CmeB protomer within the trimer is able to export drugs independently”, since you do not demonstrate drug transport. You demonstrate possible drug binding via ITC and possible H⁺ influx in the presence of CmeB, which is not present if you use CmeB_D409A. From this data, the interpretation could be “CmeB transport drugs”, if you assume it to be coupled to the observed H⁺ transport.

We have changed the statement to “Using sm-FRET imaging, we demonstrate that each CmeB protomer within the trimer is able to function independently” in this revised version of the manuscript.

16. How stable are the proteoliposomes in the presence of up to 10 μ M Taurodeoxycholate? Is there no interference with the integrity of the proteoliposomes? (Is the Δ pH affected?)

We have performed in vitro proteoliposome assays and demonstrated that the presence of 10 μ M taurodeoxycholate does not affect the integrity of the liposomes.

Referee #1: OK.

OK

17. I think another FRET pair should be tested as negative control, i.e. a pair which does not change its distance upon changing into the different conformational states (based on the structure).

The residue K781, which is located at the funnel domain of protein, was selected as a control. The results indeed confirm the robust performances of fluorophores and sm-FRET experiments.

Referee #1: Very important control. OK.

OK

18. The frequency of transitions is reduced 7 times for the D409A variant. This appears not so much for a transporter which is incapable of transporting drugs and H⁺ (again, this has to be shown in a phenotypical assay for CmeB as well, otherwise it is just an assumption). I agree that this can be interpreted that certain states still can be probed by the transporter, but why have not only less states, but also slower transitions? Do protons still have binding capacity in the transporter? Would the transitions be completely vanished if a double variant (also replacing the second Asp) would be tested?

D409A variant shows not only 7 times reduced frequency of transitions but also the longer dwelling time of L \diamond II and lack of the extrusion state. As the population level, we observed 95% of Low FRET at center of 0.2. We believe that such slow transition and without extrusion state, the D409A mutant is not able to export drug molecules. To prove this, we performed in vivo susceptibility assay to determine the function of D409A and D410A variants. The results indicate that both a single point mutation on D409 or D410 to an alanine can abolish the function of the pump. In addition, in vitro proton transport assay also shows the D409A mutant is not able to allow for the transport of protons cross the membrane. These two sets of new experimental data have been included in the revised version of this manuscript.

Referee #1: OK.

OK

19. I have a conceptual question on the description of the resting state. The resting state has its cleft closed and theoretically no access path for drugs. If I understand it correctly, the opening and conversion into a binding state would be pmf dependent. Under physiological conditions, when does this occur? Cells always have pmf (unless they

are dead), so why have a resting state? On the other hand, if drug would be a trigger (which seems at least intuitively logical), where is the drug binding site in the resting state with the closed cleft?

There are several intermediate states of the CmeB pump during the transport cycle. We have found the resting, binding and extrusion states. During drug export, the pump must go through all of these intermediate states to complete the transport cycle. We have observed the resting state of the pump both in the presence and absence of the PMF. However, it appears that this resting state is much more stable in the absence of the PMF. The conformations of the resting and extrusion forms are significantly different and can be easily to separate into two different states. It also appears that the CmeB pump may not be ready to bind drug within the periplasmic drug binding site when it is in the resting state. To bind drug molecules, the CmeB pump may need to shift to the binding state. The extrusion state allows the pump to export drug and complete the transport cycle.

Referee #1: Above is all true. But why have a resting state? Especially if the protomers cycle independently, only binding and extrusion states seem necessary. There appears no physiological need for a resting state (in the presence of a pmf). I still am puzzled by this observation, and in the light of independent cycling there seems to be no obvious interpretation.

We think that the “resting” state of the pump, in which the periplasmic cleft is closed and no channel is found in the periplasmic domain, may mark the starting point of the transport cycle. Indeed, molecular dynamics calculation in a phospholipid membrane/water environment also suggested the existence of a similar state in the AcrB pump (Fischer et al., 2013, BBA-Biomembrane, 1828:632-641). A new statement “A similar conformational state in the AcrB pump has also been found based on molecular dynamics simulations” has been added on p.7 of this revised version of the manuscript.

20. Related to this, line 221 states that one of the transitions in the absence of drug is considered “resting-resting”. Nevertheless, in the proteoliposomal FRET experiment, a Δ pH is present. If the resting state is only present in the absence of pmf, how is this possible (or I misunderstood the definition of “pmf-dependent”).

The majority transitions in the absence of drug are between resting and binding state, which correspond to the FRET observations of “resting-resting” to “resting-binding” or “resting-binding” to “resting-resting”. In fact, the resting state can be found both in the presence and absence of PMF. Since the transitions are reversible, we also observe the resting state even in the presence of PMF. However, it appears that the resting state is much more stable when there is no PMF.

Referee #1: OK (but see below).

OK

21. In lines 248-250, the statement couples the presence of pmf and substrate to be triggers for resting to binding conversion. If both are necessary, the question remains when the cell lacks the pmf in any stage of its life?

This is not the focus of our research. We would like to point out that there are several intermediate states through the transport cycle of the CmeB pump. In this case, we have identified the resting, binding and extrusion states of the pump. The alteration between different states may need to couple with the transfer of protons. The resting state is one of the intermediates that the pump must go through during drug export. It appears that the resting state of the pump is much more stable in the absence of proton transfer.

Referee #1: This might not be the focus of your research, however, you made this statement based on your observations and of course it has to translate to the physiological role of the pump states, otherwise why do the research? If the authors state that the pmf is a trigger, besides substrates, I think the question of when the pmf can be a trigger in a living cell (where the pmf is constantly present (or is it?)) is warranted. Btw. your answer above is not even trying to interpret that result, it only describes the results you obtained.

Based on our single-molecule FRET studies with the D409A mutant, we clearly observed that the CmeB pump was not able to advance its cycle from the “resting” to “binding” states in the absence of the PMF. We hypothesize that binding of substrates triggers conformation change, which advances the transport cycle from “resting” to “binding”

states. However, this process is energy dependent, which requires the PMF.

22. A technical question: Why is cyclooctatetraene (2 mM) added to the imaging experiments. Could it be a substrate of the pump as well and has this been checked?

Cyclooctatetraene (Cot) is an efficient triplet state quencher that reduces the lifetime of dark states. To address the concern that Cot could be a substrate of CmeB, we measured the binding affinity of Tdc to CmeB both in the presence and absence of 2mM Cot. The results indicate that the presence of Cot has no significant effect on the binding of Tdc to the CmeB pump.

Referee #1: OK, but it shows (although I have not seen the results of the experiment, I have to believe the statement above) only that Cot is not directly competing with Tdc. RND pumps appear to have multiple binding sites (Yu et al., 2003, Yu et al., 2005, Murakami et al., 2006, Nakashima et al., 2011, Eicher et al., 2012, Oswald et al., 2016).

The binding results were shown in Supplementary Table 5 of the last version of the manuscript. These data are also included in this version of the revised manuscript (Supplementary Table 5).

23. Lines 400-406, what was the % of the total proteoliposomes (traces) which did match with the four criteria?

Approximately 10% of molecules match these criteria.

Referee #1: OK.

OK

Reviewer #2 (Remarks to the Author):

In their revised manuscript The authors have adequately and in a satisfactory manner addressed my critique of the original article. I recommend the revised manuscript for publication in Nature Communications in its present form.

Great

Reviewer #3 (Remarks to the Author):

I am more than satisfied with the responses to my previous enquires. A great paper!

Great